# Rab5c-mediated endocytic trafficking regulates hematopoietic stem and progenitor cell development via Notch and AKT signaling

**Jian Heng**[1,2,3], **Peng Lv**[1,2,3], **Yifan Zhang**[1,2,3], **Xinjie Cheng**[1], **Lu Wang**[4], **Dongyuan Ma**[1,2], **Feng Liu**[1,2]*

1 State Key Laboratory of Membrane Biology, Institute of Zoology, Chinese Academy of Sciences, Beijing, China, 2 Institute for Stem Cell and Regeneration, Chinese Academy of Sciences, Beijing, China, 3 University of Chinese Academy of Sciences, Beijing, China, 4 State Key Laboratory of Experimental Hematology, Institute of Hematology and Blood Diseases Hospital, Chinese Academy of Medical Sciences & Peking Union Medical College, Tianjin, China

* liuf@ioz.ac.cn

**Data Availability Statement:** The relevant data are within the paper and its Supporting Information files. The original RNA-Seq data of fli1a+ cells in control and rab5c morphants have been deposited

## Abstract

It is well known that various developmental signals play diverse roles in hematopoietic stem and progenitor cell (HSPC) production; however, how these signaling pathways are orchestrated remains incompletely understood. Here, we report that Rab5c is essential for HSPC specification by endocytic trafficking of Notch and AKT signaling in zebrafish embryos. Rab5c deficiency leads to defects in HSPC production. Mechanistically, Rab5c regulates hemogenic endothelium (HE) specification by endocytic trafficking of Notch ligands and receptor. We further show that the interaction between Rab5c and Appl1 in the endosome is required for the survival of HE in the ventral wall of the dorsal aorta through AKT signaling. Interestingly, Rab5c overactivation can also lead to defects in HSPC production, which is attributed to excessive endolysosomal trafficking inducing Notch signaling defect. Taken together, our findings establish a previously unrecognized role of Rab5c-mediated endocytic trafficking in HSPC development and provide new insights into how spatiotemporal signals are orchestrated to accurately execute cell fate transition.

## Introduction

Hematopoietic stem and progenitor cells (HSPCs), which have the abilities of self-renewal and multilineage differentiation potential throughout the lifetime, hold great promise for rebuilding the hematopoietic system [1]. During vertebrate embryogenesis, the production of HSPCs takes place within the aorta-gonad-mesonephros (AGM in mammals) region or the ventral wall of the dorsal aorta (VDA in zebrafish) [2–4], where a subset of endothelial cells (ECs) gradually convert their fate into hemogenic endothelium (HE) cells and further give rise to HSPCs through endothelial-to-hematopoietic transition (EHT) [4–7]. At molecular level, the orchestration of various developmental signals ensures the implementation of cell fate transition. However, how ECs process the different developmental signals and accurately execute fate transition to HSPCs remains incompletely understood.

in the SRA database under accession number PRJNA542436.

**Funding:** The research was funded by grants from the National Key Research and Development Program of China (2018YFA0800200, and 2016YFA0100500, FL; 2018YFA0801000, DM), the Strategic Priority Research Program of the Chinese Academy of Sciences, China (XDA16010207, FL), the National Natural Science Foundation of China (31830061, 81530004, and 31425016, FL), and Youth Innovation Promotion Association, CAS (2016083, DM). The funders had no role in study design, data collection and analysis, decision to publish, or preparation of the manuscript.

**Competing interests:** The authors have declared that no competing interests exist.

**Abbreviations:** AGM, aorta-gonad-mesonephros; CA, constitutively active; CDS, coding sequence; CHT, caudal hematopoietic tissue; co-IP, co-immunoprecipitation; DA, dorsal aorta; DN, dominant-negative; dpf, days post fertilization; EC, endothelial cell; EE, early endosome; EGFR, epidermal growth factor receptor; EHT, endothelial-to-hematopoietic transition; FACS, fluorescence-activated cell sorting; FISH, fluorescence in situ hybridization; GDP, guanosine diphosphate; GFP, green fluorescent protein; GO, gene ontology; gRNA, guide RNA; GTP, guanosine triphosphate; HDBEC, human dermal blood EC; HE, hemogenic endothelium; hpf, hours post fertilization; HRM, high-resolution melting curve; HS, heat shock; HSPC, hematopoietic stem and progenitor cell; HUVEC, human umbilical vein EC; IF, immunofluorescence; KD, knockdown; MO, morpholino; NICD, Notch intracellular domain; p-Akt, phosphorylated Akt; PAM, protospacer adjacent motif; p-Erk, phosphorylated Erk; PIK3CA, catalytic subunit PI3K-alpha; qRT-PCR, quantitative reverse-transcription PCR; RNA-seq, RNA sequencing; RPKM, reads per kilobase of exon model per million mapped reads; TF, transferrin; TRITC, tetramethylrhodamine; TUNEL, terminal-deoxynucleoitidyl transferase mediated nick end labeling; VDA, ventral wall of the dorsal aorta; WISH, whole-mount in situ hybridization; WT, wild type.

Endocytic trafficking plays a crucial role in signaling specificity regulation through transporting signal molecules to a series of specialized intracellular compartments. Rab5 belongs to the small GTPases of Rab family, which is indispensable for transmembrane protein endocytosis, clathrin-coated vesicle–early endosome (EE) fusion, EE–EE fusion, EE maturation, and endosome motility [8–10]. A large scale mRNA profiling from human and mouse tissues shows distinct tissue distributions of the Rab5 isoforms [11], suggesting that the different Rab5 isoforms support physiological-biological activities through versatile endocytic trafficking. Previous reports have demonstrated that Rab5 isoforms are involved in diverse biological processes, including phagocytosis, membrane currents regulation, cell motility, and somatic cell reprogramming [12–17]. However, it remains unknown whether Rab5 isoforms play a role in transportation of developmental signals and thus affect HSPC production in vertebrate embryos.

Several signaling pathways have been shown to be essential for HSPC development; among them, Notch signaling is involved in embryonic HSPC production through cellular communication and signal transduction [18,19]. The transmembrane Notch ligands Jagged and Delta families interact with Notch receptors, causing a proteolytic cleavage in the Notch extracellular domain and further facilitating γ-secretase-regulated Notch intracellular domain (NICD) release [20,21]. Then NICD enters the nucleus to activate target gene transcription [22–24]. The Notch direct and indirect downstream genes, such as *hey2*, *gata2b* and *runx1*, are required for HE specification [25–29]. Studies on *mind bomb* and dynamin-regulated Delta endocytosis [30,31] and monoubiquitination-mediated *trans*-endocytosis of Notch receptor [32,33] imply that not only the Notch ligands but also the receptors are tightly associated with endocytic trafficking to function properly. In addition, AKT signaling is another important regulator for HSPC production through supporting cell survival [34–36], and endocytic trafficking is also involved in AKT signaling transduction [37,38]. However, how these spatiotemporal signaling pathways are orchestrated in the hematopoietic region still remains elusive.

Here, we have demonstrated that Rab5c-mediated endocytic trafficking is crucial for HSPC production. Rab5c deficiency leads to defects in HE specification and survival. Further mechanistic studies demonstrate that endocytic trafficking regulated by Rab5c is required for Notch and AKT signaling transduction, which are indispensable for HSPC production. Additionally, Rab5c overactivation leading to excessive endolysosomal trafficking can also cause defects in HSPC production, suggesting that a relatively balanced endocytic trafficking ensures normal HSPC development during embryogenesis.

## Results

### Endocytic trafficking–related pathways and their regulator Rab5c are enriched in zebrafish hematopoiesis tissue during early development

Previously, in order to find potential definitive hematopoiesis regulators, we sorted ECs ($kdrl^+runx1^-$) and HE cells ($kdrl^+runx1^+$) from the trunk region of *Tg*(*kdrl*:mCherry/*runx1*:en-GFP) double transgenic line for RNA sequencing (RNA-seq) (Fig 1A) [39]. Interestingly, gene ontology (GO) analysis showed that up-regulated transcripts in HE cells were enriched in the endocytic trafficking related terms, such as endosome, vesicle-mediated transport, and endocytosis (Fig 1B). Notably, the expression of an important endocytic trafficking regulator, *rab5c*, was also enriched in HE cells and was the highest one among the genes of the Rab5 family in ECs and HE cells (Fig 1C). To further confirm the expression pattern of *rab5c*, whole-mount in situ hybridization (WISH) and double fluorescence in situ hybridization (FISH) were carried out. The results showed that expression of *rab5c* was highly enriched in the VDA region and particularly, in *fli1a*+ cells (including ECs and HE cells) at 26 hours post

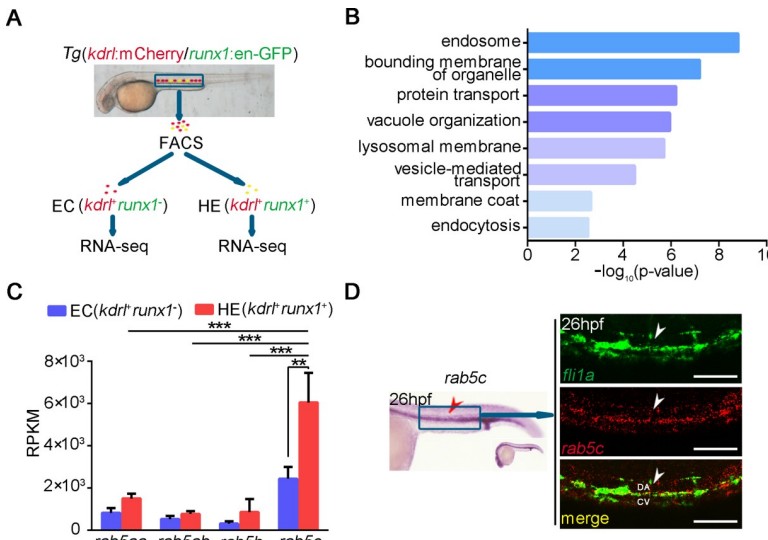

**Fig 1. Endocytic trafficking regulator Rab5c is highly expressed in embryonic hematopoiesis region.** (A) Flowchart of FACS and RNA-seq analysis of the different types of cells in the trunk region from *Tg(kdrl*:mCherry/*runx1*:en-GFP) transgenic zebrafish embryos at about 28 hpf. (B) GO analysis of the sequencing data shows that up-regulated transcripts in *runx1*⁺ HE are enriched in the endosome and endocytic trafficking related terms. (C) RPKM values of Rab5 family genes in EC and HE. (D) Expression pattern of *rab5c* mRNA examined by WISH and double FISH. Scale bar, 200 μm. The *P* values in this figure were calculated by Student *t* test. The underlying data in this figure can be found in S1 Data. CA, caudal vein; DA, dorsal aorta; EC, endothelial cell; FACS, fluorescence-activated cell sorting; FISH, fluorescence in situ hybridization; GO, gene ontology; HE, hemogenic endothelium; hpf, hours post fertilization; RNA-seq, RNA sequencing; RPKM, reads per kilobase of exon model per million mapped reads; WISH, whole-mount in situ hybridization.

fertilization (hpf; Fig 1D). Quantitative reverse-transcription PCR (qRT-PCR) analysis of cells sorted from trunk region of *Tg(fli1a*:EGFP) also indicated that expression of *rab5c* was enriched in *fli1a*⁺ cells (S1A and S1B Fig). Taken together, these results showed that genes involved in endocytic trafficking related pathways and, in particular, *rab5c*, are enriched in definitive hematopoiesis tissue, suggesting their possible roles in HSPC development.

## Rab5c is required for HSPC development

To investigate whether Rab5c is required for zebrafish HSPC production, we used a knockdown (KD) approach by *rab5c* anti-sense ATG morpholino (MO) injection (S2A Fig), and the MO effectiveness in translation-blocking was verified by western blot (S2B and S2C Fig). WISH showed that the primitive hematopoiesis was normal (S2D Fig), and time-lapse imaging showed that blood flow was not altered (S1 and S2 Movies), whereas HSPC production in definitive hematopoiesis was seriously decreased in *rab5c* morphants (MO oligonucleotides injected embryos) compared with control (Fig 2A and 2B). Importantly, overexpression of *rab5c* mRNA lacking the MO binding site, which was not inhibited by *rab5c* MO (S2E and S2F Fig), could rescue this HSPC defect (S2G Fig), suggesting a gene-specific phenotype. Rab5c-KD induced HSPC defect further led to a reduced number of HSPC derivatives, including erythroid, myeloid, and lymphoid cells at 4 days post fertilization (dpf; Fig 2C–2E). We observed definitive hematopoietic precursors by using the *Tg(kdrl*:mCherry/*cmyb*:GFP) double transgenic line and found that the number of precursors (*kdrl*⁺*cmyb*⁺) within the VDA region was markedly decreased in *rab5c* morphants (Fig 2F and 2G). To validate the HSPC phenotype upon Rab5c KD, we used a well-established HSPC transgenic line, *Tg(CD41*:GFP), to observe living HSPCs in the caudal hematopoietic tissue (CHT) at 2 dpf and found that the number of

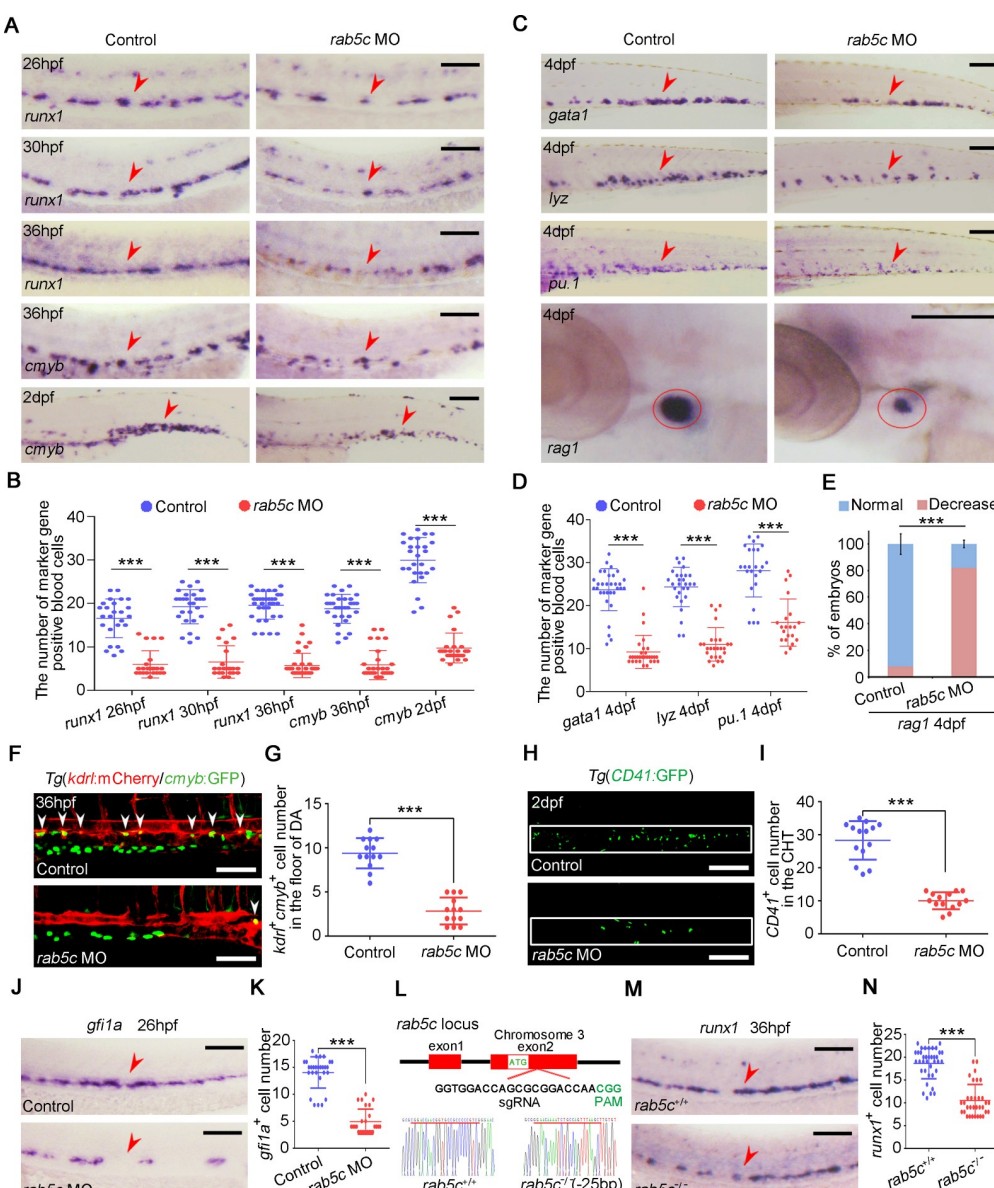

**Fig 2. Rab5c-deficiency impairs HSPC production.** (A) Expression of HSPC markers *runx1* and *cmyb* in control and *rab5c* morphants examined by WISH. The red arrowheads denote HSPCs. Scale bar, 100 μm. (B) Quantification of WISH results. Error bars, mean ± SD, ***P < 0.001. (C) Expression of differentiation markers: erythroid marker *gata1*, neutrophil marker *lyz*, myeloid marker *pu.1*, and lymphoid marker *rag1* in the control and *rab5c* morphants at 4 dpf examined by WISH. Scale bar, 100 μm. (D) Quantification of panel C. Error bars, mean ± SD, ***P < 0.001. (E) Statistical analysis of the WISH. Error bars, mean ± SD, ***P < 0.001. (F) Confocal imaging shows the *kdrl*+*cmyb*+ definitive hematopoietic precursors in the VDA region of control and *rab5c* morphants at 36 hpf. White arrowheads denote precursors. Scale bar, 100 μm. (G) Quantification of *kdrl*+*cmyb*+ precursors. Error bars, mean ± SD, ***P < 0.001. (H) Confocal imaging shows that there are less *CD41*+ HSPCs (green) in the CHT (box area) of *rab5c* morphants compared with control at 2 dpf. Scale bar, 100 μm. (I) Quantification of *CD41*+ HSPCs. Error bars, mean ± SD, ***P < 0.001. (J) Expression of HE marker *gfi1a* in control and *rab5c* morphants examined by WISH. Scale bar, 100 μm. (K) Quantification of the *gfi1a* positive cells. Error bars, mean ± SD, ***P < 0.001. (L) Generation of *rab5c* frameshift mutant using the CRISPR/Cas9 technique. Location and sequence of the target site are exhibited. *rab5c* wild type and mutant sequences are listed below. (M) Expression of *runx1* in *rab5c* wild type and mutant at 36 hpf examined by WISH. Scale bar, 100 μm. (N) Quantification of the *runx1* positive cells. Error bars, mean ± SD, ***P < 0.001. The *P* values in this figure were calculated by Student *t* test. The underlying data in this figure can be found in S1 Data. CHT, caudal hematopoietic tissue; dpf, days post fertilization; DA, dorsal aorta; HE, hemogenic endothelium; HSPC, hematopoietic stem and progenitor cell; hpf, hours post fertilization; PAM, protospacer adjacent motif; VDA, ventral wall of the dorsal aorta; WISH, whole-mount in situ hybridization .

$CD41^+$ HSPCs was severely reduced in *rab5c* morphants (Fig 2H and 2I). Furthermore, we examined the HE specification by the marker *gfi1a* and found that *gfi1a* expression was decreased in *rab5c* morphants (Fig 2J and 2K), indicating that Rab5c is required for HE specification. To examine whether Rab5c is also required for the subsequent EHT process, we performed time-lapse imaging. The results showed that the $kdrl^+cmyb^+$ cell number in *rab5c* morphants was much less than that in the control (S3 and S4 Movies). However, the few $kdrl^+cmyb^+$ cells in *rab5c* morphants were able to transform to spherical shape and emerged from the DA (S4 Movie, S2H Fig), which mimicked the $kdrl^+cmyb^+$ cells in the control group (S3 Movie). These results suggest that Rab5c is indispensable for HE specification but dispensable for EHT.

To further demonstrate the Rab5c loss-of-function phenotype above, we generated a Rab5c-null frameshift mutant by CRISPR/Cas9 technology (Fig 2L). WISH showed that the HSPC production was reduced in *rab5c* mutant (Fig 2M and 2N). Notably, the HSPC phenotype in *rab5c* mutant was relatively weaker than that in *rab5c* morphants. We wondered whether homologous gene compensation occurred in the *rab5c* mutant, which can lead to alleviative phenotype [40–44]. Then we examined the expression level of other members of zebrafish Rab5 family and found that *rab5ab* and *rab5b*, but not *rab5aa*, were obviously increased in *rab5c* mutant (S2I Fig). To suppress the genetic compensation, we knocked down *rab5ab* and *rab5b* in *rab5c* mutant by co-injection of low-dose of *rab5ab* and *rab5b* MOs. WISH showed that low-dose of MOs co-injection in *rab5c* mutant, but not in wild-type, led to severe HSPC defect (S2J Fig), which fully phenocopied *rab5c* morphants. We also generated *rab5ab* and *rab5b* single-gene frameshift mutants, respectively, using CRISPR/Cas9 (S2K and S2L Fig). WISH showed that *rab5ab* and *rab5b* mutants had relatively normal *runx1* expression (S2M and S2N Fig). No homologous gene compensation effect was observed in either *rab5ab* or *rab5b* mutants (S2O and S2P Fig). To obtain the *rab5ab/rab5c* and *rab5b/rab5c* double-knockout mutants, we crossed *rab5ab* or *rab5b* mutant with *rab5c* mutant, respectively. WISH showed that *runx1* in both *rab5ab/rab5c* and *rab5b/rab5c* double-knockout mutants was severely decreased (S2Q and S2R Fig), and the double-knockout fully phenocopied the *rab5c* MO-mediated KD.

## Rab5c is indispensable for HE specification in an EC autonomous manner

To further demonstrate the role of Rab5c in HSPC development, the dominant-negative (DN) form of zebrafish Rab5c was generated by a S36N amino acid mutation in the conserved guanosine triphosphate (GTP)-binding pocket (Fig 3A), which affects GTP/guanosine diphosphate (GDP) affinity [45–47], and its effect on endocytic trafficking inhibition was confirmed by internalization assay in Hela cell line (S3A and S3B Fig). We also observed green fluorescent protein (GFP)-tagged Rab5c DN and Rab5c wild type (WT) in Hela cells and found that Rab5c DN was not able to aggregate in the characteristics of endosomal pattern (S6 Movie) as Rab5c WT did (S5 Movie), further confirming the functional properties of Rab5c. To determine the timing of Rab5c function, we utilized *hsp70*-mediated heat-shock (HS) inducible Rab5c DN overexpression system. Strong GFP-Rab5c DN expression was detected from 2 hours post HS at 20 hpf to at least 20 hours post HS (S3C Fig). WISH showed that expression of HSPC marker *cmyb* was moderately decreased in group of HS induction of Rab5c DN at 20 hpf (Fig 3B and 3C). To demonstrate whether Rab5c is required EC autonomously for HSPC development, a construct expressing GFP-tagged Rab5c DN driven by the *fli1a* promoter was generated, and the GFP signal was detected in the ECs of *fli1a*-GFP-*rab5c* DN injected embryos (S3D Fig). WISH showed that EC-specific overexpression of Rab5c DN led to HSPC defects (Fig 3D and 3E). As a control, we made a new construct for somite-specific overexpression of

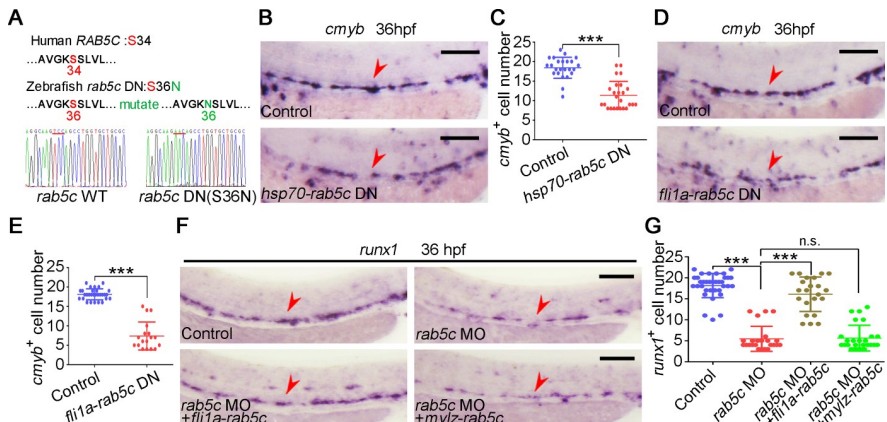

**Fig 3. Rab5c function is indispensable for HE specification in an EC autonomous manner.** (A) Amino acid sequence is conserved for GTP-binding pockets of the Rab5c proteins between human and zebrafish. The amino acid of zebrafish in red was mutated to generate DN Rab protein by affecting GTP/GDP affinity. (B) Expression of *cmyb* in control and Rab5c inhibition groups examined by WISH. *rab5c* DN overexpression was carried out by *hsp70*-GFP-*rab5c* DN HS at 20 hpf. Scale bar, 100 μm. (C) Quantification of the *cmyb* positive cells. Error bars, mean ± SD, ***$P < 0.001$. (D) WISH analysis shows that *cmyb* is decreased in embryos of *rab5c* DN overexpression driven by a *fli1a* promoter. Scale bar, 100 μm. (E) Quantification of the *cmyb* positive cells. Error bars, mean ± SD, ***$P < 0.001$. (F) Endothelial, but not somitic, Rab5c overexpression has a HSPC rescue. Scale bar, 100 μm. (G) Quantification of the *runx1* positive cells. Error bars, mean ± SD, ***$P < 0.001$. The *P* values in this figure were calculated by Student *t* test. The underlying data in this figure can be found in S1 Data. DN, dominant-negative; EC, endothelial cell; GDP, guanosine diphosphate; GTP, guanosine triphosphate; HE, hemogenic endothelium; hpf, hours post fertilization; HS, heat shock; HSPC, hematopoietic stem and progenitor cell; n.s., nonsignificant; WISH, whole-mount in situ hybridization; WT, wild type .

*rab5c* mRNA lacking the MO binding site by *mylz* promoter (S3E Fig). We found that only the endothelial but not the somitic overexpression of Rab5c had a rescue effect (Fig 3F and 3G), confirming its EC-specific role. To further determine whether Rab5c-deficiency impairs somitic wave of HSPC specification [48–50], we examined the expression of sclerotome markers by WISH. The results showed that the expression of *dlc*, *vegfa*, and *shh* was not changed in Rab5c-dificiency embryos at 16 hpf (S3F and S3G Fig), suggesting Rab5c not involved in somitic wave of HSPC specification. Taken together, these results indicated that Rab5c function was indispensable during the critical phase of HE specification in an EC autonomous manner.

## Rab5c-regulated endocytic trafficking is important for HE specification through Notch signaling

To identify the underlying molecular mechanisms upon Rab5c deficiency, we sorted the *fli1a*⁺ ECs from the trunk region of control and *rab5c* morphants at 26 hpf for RNA-seq. GO analysis revealed that down-regulated and up-regulated genes in Rab5c-deficiency embryos were enriched in various signaling pathways (Fig 4A and 4B). Among these signaling pathways, Notch signaling has been demonstrated to regulate HE specification [18,26,39,51,52]. Then we examined Notch signaling in control and Rab5c deficiency embryos in detail. Western blot results showed that the protein level of NICD was severely decreased in *rab5c* morphants (Fig 4C and 4D). To further demonstrate this, we next took advantage of the Notch reporter transgenic line *Tg(fli1a*:EGFP/*tp1*:mCherry) to examine Notch activity in ECs and found that the *fli1a*⁺*tp1*⁺ double positive ECs in the VDA region were markedly reduced, indicating that Notch activity was attenuated in *rab5c* morphants at 26 hpf (Fig 4E and 4F). The expression of Notch signaling downstream genes *ephrinB2a*, *gata2b*, *hey1*, and *hey2* was also decreased in *rab5c* morphants at 26 hpf (Fig 4G–4J). By contrast, the expression level of panvascular marker

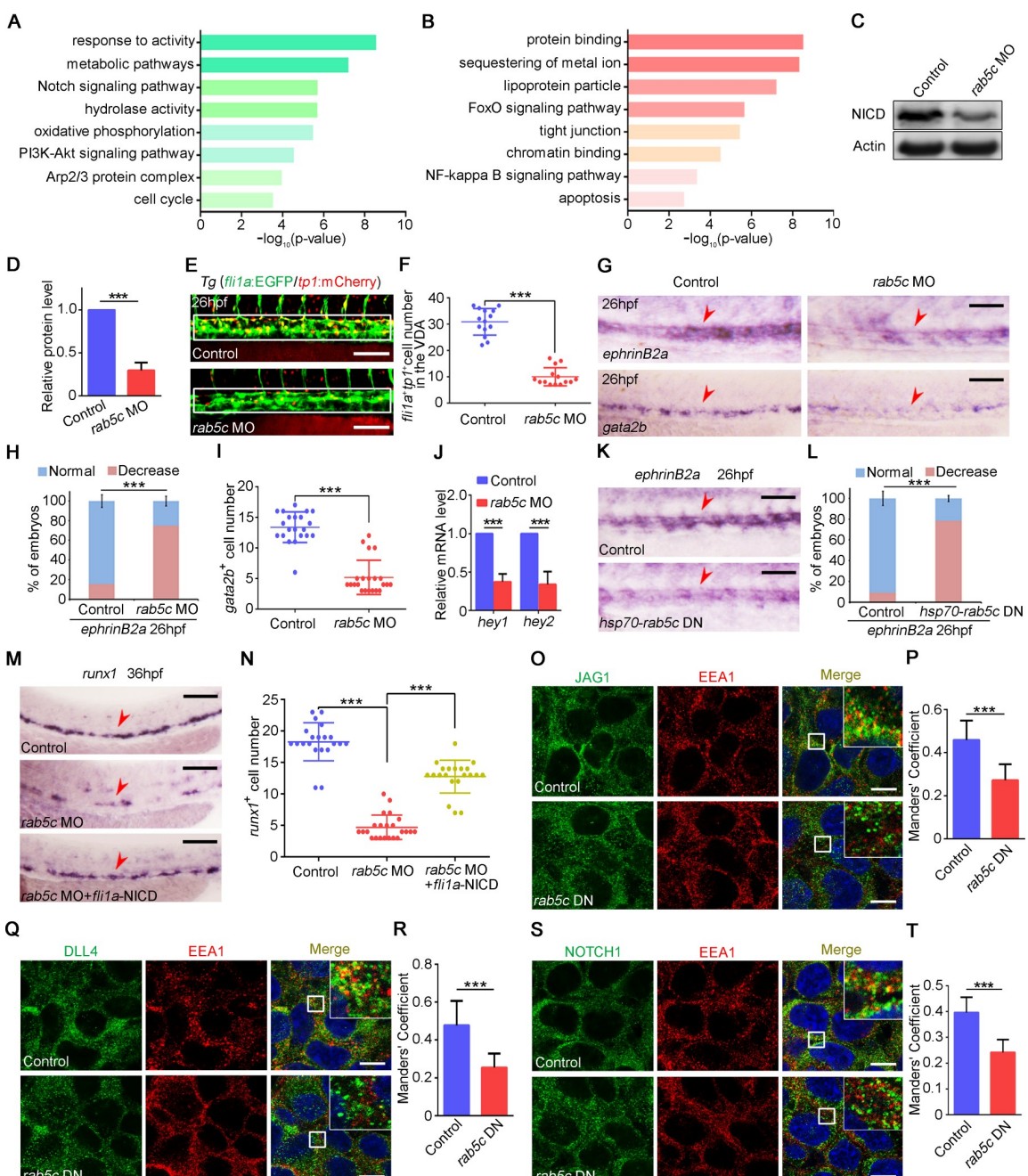

**Fig 4. Rab5c-regulated endocytic trafficking is important for HE specification through Notch signaling.** (A) The *fli1a*⁺ ECs were sorted from the trunk region of the control and the *rab5c* morphants at 26 hpf for RNA-seq. The GO analysis showing the enrichment of down-regulated signaling pathways in *rab5c* morphants. (B) GO analysis showing the enrichment of up-regulated signaling pathways in *rab5c* morphants. (C) Protein level of NICD in the control and the *rab5c* morphants at 26 hpf examined by WB. (D) Quantification and statistical analysis of WB. Quantification of protein level using gray analysis (Gel-Pro analyzer). Error bars, mean ± SD, ***$P < 0.001$. (E) Confocal imaging shows the *fli1a*⁺*tp1*⁺ Notch-active ECs in the VDA region (box area) of the control and the *rab5c* morphants. Scale bar, 100 μm. (F) Quantification of *fli1a*⁺*tp1*⁺ Notch-active ECs. Error bars, mean ± SD, ***$P < 0.001$. (G) Expression of Notch downstream genes *ephrinB2a* and *gata2b* in the control and the *rab5c* morphants at 26 hpf examined by WISH. Scale bar, 100 μm. (H) Statistical analysis of the WISH. Error bars, mean ± SD, ***$P < 0.001$. (I) Quantification of the *gata2b* positive cells. Error bars, mean ± SD, ***$P < 0.001$. (J) Relative mRNA level of Notch signaling downstream genes *hey1*, *hey2* in the control and the *rab5c* morphants at 26 hpf examined by qRT-PCR. Error bars, mean ± SD, ***$P < 0.001$. (K) Expression of *ephrinB2a* in control and Rab5c inhibition group examined by WISH. *rab5c* DN overexpression was carried out by *hsp70*-GFP-*rab5c* DN HS at 20 hpf. Scale bar, 100 μm. (L) Statistical analysis of the WISH. Error bars, mean ± SD, ***$P < 0.001$. (M) WISH shows that *runx1* expression in *rab5c* morphants is partially rescued by NICD overexpression through *fli1a*-NICD. Scale bar, 100 μm. (N) Quantification of the *runx1*

positive cells. Error bars, mean ± SD, ***$P < 0.001$. (O) Control plasmid or pCS2-*rab5c* DN transfected 293T cells were immunostained with antibodies against endogenous JAG1 (green) and EEA1 (red). Scale bar, 10 μm. (P) Quantification of co-localization of JAG1 with EEA1 using Manders' coefficient (ImageJ). $n = 14$ cells. Error bars, mean ± SD, ***$P < 0.001$. (Q) Control plasmid or pCS2-*rab5c* DN transfected 293T cells were immunostained with antibodies against endogenous DLL4 (green) and EEA1 (red). Scale bar, 10 μm. (R) Quantification of co-localization of DLL4 with EEA1 using Manders' coefficient. $n = 14$ cells. Error bars, mean ± SD, ***$P < 0.001$. (S) Control plasmid or pCS2-*rab5c* DN transfected 293T cells were immunostained with antibodies against endogenous NOTCH1 (green) and EEA1 (red). Scale bar, 10 μm. (T) Quantification of co-localization of NOTCH1 with EEA1 using Manders' coefficient. $n = 14$ cells. Error bars, mean ± SD, ***$P < 0.001$. The *P* values in this figure were calculated by Student *t* test. The underlying data in this figure can be found in S1 Data. DN, dominant-negative; EC, endothelial cell; GO, gene ontology; HE, hemogenic endothelium; hpf, hours post fertilization; HS, heat shock; NICD, Notch intracellular domain; qRT-PCR, quantitative reverse-transcription PCR; RNA-seq, RNA sequencing; VDA, ventral wall of the dorsal aorta; WB, western blot; WISH, whole-mount in situ hybridization.

*flk1* and Notch ligand *dll4* was not obviously changed (S4A Fig). We further examined the expression of Notch-independent arterial marker *tbx20* [53,54]. WISH results showed that *tbx20* expression was normal in *rab5c* morphants (S4A Fig), suggesting that the DA, where HSPCs arise, was specified normally. Our results support the view [26,55] that Notch signaling alteration does not necessarily result in DA specification defect. In addition, HS induction of Rab5c DN was carried out, and the result showed that *ephrinB2a* expression was decreased in group of Rab5c DN HS overexpression (Fig 4K and 4L). To determine whether Notch signaling induction can rescue the HSPC phenotype in *rab5c* morphants, we applied systems for EC-specific or HS inducible overexpression of NICD, which is a dominant activator of the Notch pathway. The NICD function was confirmed by overexpression of NICD alone in WT embryos and examination of its effect on endothelial and HSPC compartment. qRT-PCR showed that NICD overexpression led to up-regulation of Notch downstream genes *hey1* and *hey2* (S4B Fig), and WISH showed that NICD overexpression led to hyperproduction of *ephrinB2a* positive ECs and *cmyb*-marked HSPCs (S4C Fig). Then we performed NICD overexpression in *rab5c* morphants. WISH showed that the HSPC defect in *rab5c* morphants could be partially rescued by EC-specific induction of NICD or HS induction of NICD at 20 hpf (Fig 4M and 4N, S4D Fig). The Notch signaling in HSPC development can be divided into 2 waves: the early somitic Notch signaling (15–17 hpf) and late endothelial Notch signaling (after 20 hpf) [50,56]. It has been reported that HS induction of NICD at 14 hpf could rescue HSPC defects induced by repression of early somitic Notch signaling; however, HS just 2 hours later (about 16 hpf) cannot [50,56]. We performed early NICD induction by *hsp70*-NICD-2a-GFP HS at 14 hpf. Strong GFP expression was detected from 2 hours post HS at 14 hpf (S4E Fig) to at least 20 hours post HS. Notably, the GFP expression level at 20 hpf or 26 hpf was comparable to that at 16 hpf (S4E Fig). The WISH results showed that the HSPC defect in Rab5c-dificiency embryos could be rescued by the HS induction of NICD at 14 hpf (S4F and S4G Fig). Although HS induction of Notch at 14 hpf provided a rescue, this NICD induction was sustained throughout the endothelial wave (after 20 hpf). Together with that the HS induction of NICD at 20 hpf rescued the HSPC defect, we propose that Rab5c regulates HSPC development more likely at the endothelial wave. In addition, *rab5c* was not specifically expressed at 14 hpf or 16 hpf but was specifically expressed in ECs at 20 hpf or 26 hpf (Fig 1D, S1B and S4H Figs), further supporting that in Rab5c-dificiency embryos, Notch signaling inhibition led to the HSPC defect during the endothelial wave of HE specification.

Endocytic trafficking plays an essential role in the intramembranous cleavage and NICD release in Notch activation process [30,32,57]. We asked whether the Notch signaling defect caused by Rab5c inhibition was due to impaired endocytic trafficking. To address this question, we tried to detect the protein localization in control and Rab5c DN overexpression 293T cells by immunofluorescence (IF). IF showed that not only the co-localization of Notch ligands JAG1 and DLL4 with EE marker EEA1 but also the co-localization of Notch receptor

NOTCH1 with EEA1 were severely decreased in Rab5c DN overexpressed cells (Fig 4O–4T). These results indicated that Rab5c inhibition led to impaired endocytic trafficking of both Notch ligands and receptor, which is likely responsible for the inactivated Notch signaling in HSPC development. Previous studies showed that Rab5 family genes also affect endolysosomal trafficking, in which the early endosomes translocate to lysosomes for cargo degradation [17,58,59]. Then we examined the co-localization of Notch ligands and receptor with lysosome marker LAMP1 and found that the co-localization of JAG1, DLL4, and Notch1 with LAMP1 in Rab5c DN overexpressed cells was slightly decreased (S4I–S4N Fig), suggesting that a mild endolysosomal trafficking alteration occurred in Rab5c inhibition cells. Furthermore, we examined the total protein level of Notch ligands and receptor by WB and found that JAG1, DLL4, and NOTCH1 were normal in Rab5c DN overexpression group (S4O and S4P Fig). These results indicated that Rab5c inhibition might not affect the total protein level but specifically led to defective endocytic trafficking of Notch ligands and receptor. Taken together, we speculate that Rab5c is important for HE specification likely through endocytic trafficking-regulated Notch signaling.

## Rab5c promotes HE survival through AKT signaling

Notably, AKT signaling, an important regulator for supporting HSPC survival [34–36], was down-regulated in Rab5c-deficiency embryos in RNA-seq data (Fig 4A), and conversely, the FoxO- and apoptosis-related signaling pathways, which could be suppressed by AKT signaling, were up-regulated (Fig 4B). To determine how Rab5c-deficiency leads to AKT signaling attenuation, we firstly examined the protein level of total Akt and phosphorylated Akt (p-Akt) upon Rab5c KD. WB results showed that in *rab5c* morphants the protein level of total Akt was not changed, whereas the p-Akt was markedly decreased (Fig 5A and 5B). Consistently, Rab5c DN overexpression driven by *hsp70* promoter led to similar results (S5A and S5B Fig). To examine the survival status of HE cells within the VDA region upon Rab5c KD, we performed terminal-deoxynucleoitidyl transferase mediated nick end labeling (TUNEL) assay in control and *rab5c* MO injected *Tg*(*gfi1*:GFP) embryos, and the results showed that there were more apoptotic *gfi1*+ cells in *rab5c* morphants, in spite of that there were much fewer *gfi1*+ HE cells in *rab5c* morphants (Fig 5C and 5D). In addition, *tp53* KD mildly reduced HE cell apoptosis (Fig 5C and 5D). Furthermore, we tried to rescue the HSPCs in *rab5c* morphants through overexpression of a constitutively active (CA) form of AKT2 [60,61]. WISH showed that AKT2 CA overexpression could partially rescue *runx1* expression in *rab5c* morphants (Fig 5E and 5F). Importantly, co-overexpression of both NICD and AKT2 CA showed a more efficient rescue effect than NICD overexpression alone (Fig 5G and 5H). Previous studies demonstrate that AKT signaling is also dependent on endosome [38,62–64]. Rab5 orchestrates endocytic trafficking of Notch signaling molecules through the EE harboring its effector EEA1; however, there is another type of endosome labeled by the Rab5 effector Appl1 [37,64–66]. Rab5-bound Appl1 facilitates AKT phosphorylation through Appl1 positive endosome, which may increase the interaction between AKT and PI3K catalytic subunit PI3K-alpha (PIK3CA) [64]. The IF results in 293T cells showed that AKT preferred to co-localize with APPL1, and NOTCH1 mainly co-localized with EEA1 (S5C–S5F Fig). In order to confirm the interaction between Rab5c and Appl1, we co-expressed mCherry-Appl1 and GFP-Rab5c in 293T cell line. Confocal imaging showed that mCherry-Appl1 indeed co-localized with GFP-Rab5c in a characteristic of endosomal pattern (Fig 5I). The co-immunoprecipitation (co-IP) results showed that Appl1-Myc was co-precipitated by Flag-Rab5c, further confirming the interaction between Appl1 and Rab5c (Fig 5J and 5K). Then we examined the co-localization of AKT with APPL1 or PI3KCA in the control and Rab5c inhibition 293T cells by IF. The results showed that not

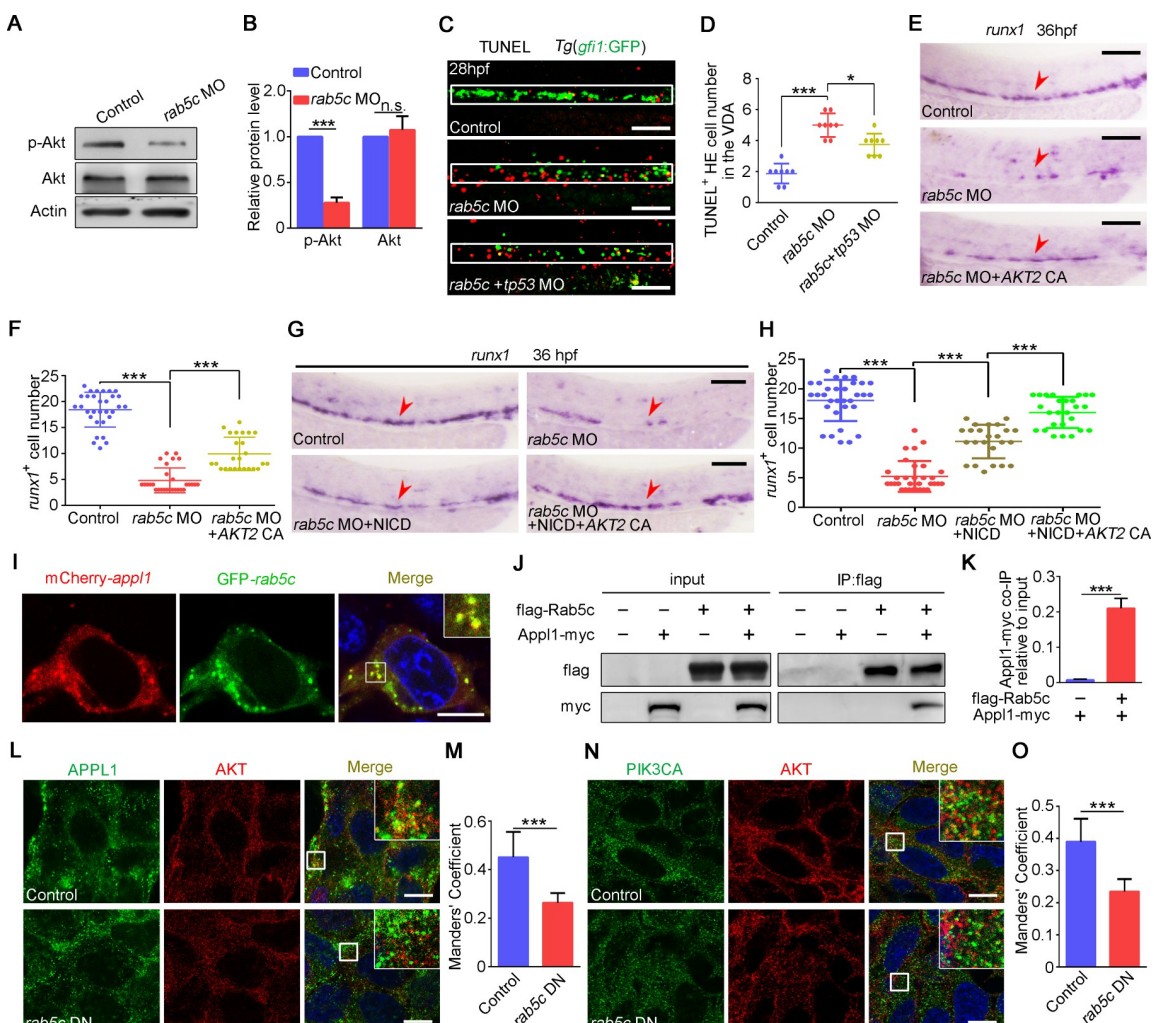

**Fig 5. Rab5c is required for HE survival through AKT signaling.** (A) Protein level of p-Akt and total Akt in the control and the *rab5c* morphants examined by WB. (B) Quantification of protein level using gray analysis (Gel-Pro analyzer). Error bars, mean ± SD, $^{***}P < 0.001$. (C) TUNEL assay shows that there are more apoptotic HE cells (yellow) in the VDA region (box area) of *rab5c* morphants compared with the control. Injection of *tp53* MO just slightly reduces apoptotic HE cell number in *rab5c* morphants. Scale bar, 100 μm. (D) Quantification of TUNEL$^+$ HE cells. Error bars, mean ± SD, $^{*}P < 0.05$, $^{***}P < 0.001$. (E) HSPC rescue of *rab5c* morphants with CA form *AKT2* mRNA. Scale bar, 100 μm. (F) Quantification of the *runx1* positive cells. Error bars, mean ± SD, $^{***}P < 0.001$. (G) Co-overexpression of NICD and *AKT2* CA shows a more efficient rescue effect than NICD overexpression alone. Scale bar, 100 μm. (H) Quantification of the *runx1* positive cells. Error bars, mean ± SD, $^{***}P < 0.001$. (I) Confocal imaging shows partial co-localization of Appl1 and Rab5c in 293T cells co-transfected with pCS2-mCherry-*appl1* and pCS2-GFP-*rab5c* constructs. Scale bar, 10 μm. (J) 293T cells were co-transfected with flag-tagged *rab5c* and myc-tagged *appl1* constructs. Cell lysate was subjected to IP using anti-flag beads followed by WB analysis. (K) Quantification of protein level using gray analysis (Gel-Pro analyzer). The ratios of Appl1-myc co-IP relative to input were calculated. Error bars, mean ± SD, $^{***}P < 0.001$. (L) Control plasmid or pCS2-*rab5c* DN transfected 293T cells were immunostained with antibodies against endogenous APPL1 (green) and AKT (red). Scale bar, 10 μm. (M) Quantification of co-localization of APPL1 with AKT using Manders' coefficient (ImageJ). $n = 14$ cells. Error bars, mean ± SD, $^{***}P < 0.001$. (N) Control plasmid or pCS2-*rab5c* DN transfected 293T cells were immunostained with antibodies against endogenous PIK3CA (green) and AKT (red). Scale bar, 10 μm. (O) Quantification of co-localization of PIK3CA with AKT using Manders' coefficient. $n = 14$ cells. Error bars, mean ± SD, $^{***}P < 0.001$. The $P$ values in this figure were calculated by Student $t$ test. The underlying data in this figure can be found in S1 Data. CA, constitutively active; DN, dominant-negative; GFP, green fluorescent protein; HE, hemogenic endothelium; HSPC, hematopoietic stem and progenitor cell; IP, immunoprecipitation; MO, morpholino; NICD, Notch intracellular domain; n.s., nonsignificant; p-Akt, phosphorylated Akt; PIK3CA, catalytic subunit PI3K-alpha; TUNEL, terminal-deoxynucleoitidyl transferase mediated nick end labeling; VDA, ventral wall of the dorsal aorta; WB, western blot .

only the co-localization of APPL1 with AKT (Fig 5L and 5M) but also the co-localization of PIK3CA with AKT (Fig 5N and 5O) were severely decreased in Rab5c DN overexpressed cells. These results suggested that Rab5c is essential for AKT signaling activation through Appl1 positive endosome. Interestingly, Appl1 KD also led to similar HSPC defect (S5G Fig). Taken together, these results suggested that Rab5c is important for promoting HE survival via Appl1-mediated AKT signaling.

Furthermore, we sought to determine whether other pathways were affected by Rab5c-deficiency. Among these pathways, Wnt and Erk signals are known to be important for HSPC production [67–69]. We examined the Wnt signaling by using Wnt activity reporter *Tg*(TOPflash: GFP) line and found that Rab5c KD did not cause Wnt activity alteration (S5H Fig). In addition, the Wnt signaling downstream gene *c-myc*, *mycbp*, and *cldn1* expression did not change in *rab5c* morphants (S5I Fig). Next we checked whether the Erk signaling was affected in *rab5c* morphants. WB for examination of total Erk and phosphorylated Erk (p-Erk) protein level was carried out as previously reported [68,69]. The results showed that the protein level of total Erk and p-Erk was not obviously changed in *rab5c* morphants compared with control (S5J and S5K Fig). Previous studies in human cell line reported that suppression of Rab5a affected epidermal growth factor receptor (EGFR), whereas Rab5c suppression had very little effect [70,71]. Considering that Erk is the EGFR downstream gene, unchanged Erk signaling in *rab5c* morphants in our work suggests that Rab5c is not essential for EGFR function during HSPC development. Finally, previous studies show that Vegf acts downstream of Shh and upstream of the Notch signaling to facilitate HSPC production [48,49]. We performed WISH to examine whether Shh and Vegf were impaired in Rab5c-deficiency embryos. The results showed that the expression of *shh* and *vegfa* at 16 or 26 hpf was not changed in *rab5c*

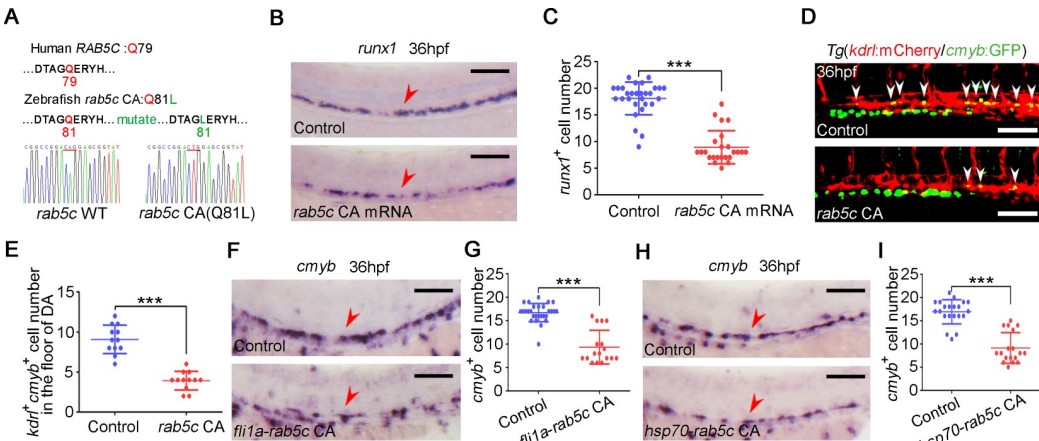

**Fig 6. Rab5c overactivation leads to HSPC production defect.** (A) Amino acid sequence is conserved for GTP hydrolysis of the Rab5c proteins between human and zebrafish. The amino acid of zebrafish in red was mutated to generate CA Rab protein by affecting GTPase activity. (B) WISH analysis shows that *runx1* is decreased in *rab5c* CA mRNA overexpressed embryos. Scale bar, 100 μm. (C) Quantification of the *runx1* positive cells. Error bars, mean ± SD, ***P < 0.001. (D) Confocal imaging shows the *kdrl*+*cmyb*+ definitive hematopoietic precursors in the VDA region of control and Rab5c CA overexpression group at 36 hpf. White arrowheads denote precursors. Scale bar, 100 μm. (E) Quantification of *kdrl*+*cmyb*+ cells. Error bars, mean ± SD, ***P < 0.001. (F) WISH analysis shows that *cmyb* is decreased in embryos of *rab5c* CA overexpression driven by *fli1a* promoter. Scale bar, 100 μm. (G) Quantification of the *cmyb* positive cells. Error bars, mean ± SD, ***P < 0.001. (H) Expression of *cmyb* in control and Rab5c CA overexpression group examined by WISH. *hsp70*-GFP-*rab5c* CA construct injected embryos were HS at 20 hpf for Rab5c CA overexpression. Scale bar, 100 μm. (I) Quantification of the *cmyb* positive cells. Error bars, mean ± SD, ***P < 0.001. The *P* values in this figure were calculated by Student *t* test. The underlying data in this figure can be found in S1 Data. CA, constitutively active; GFP, green fluorescent protein; GTP, guanosine triphosphate; hpf, hours post fertilization; HS, heat shock; HSPC, hematopoietic stem and progenitor cell; VDA, ventral wall of the dorsal aorta; WISH, whole-mount in situ hybridization .

morphants compared with the control (S3F and S3G Fig, S5L–S5N Fig), suggesting that Shh and Vegf pathways might not be involved in Rab5c-dificiency induced HSPC defect. Taken together, these results suggest that Rab5c-dificiency induced HSPC defect is probably not through Wnt, Erk, and Vegf and Shh signaling.

## Rab5c overactivation leads to HSPC production defect

Because we have demonstrated that Rab5c inhibition impaired HSPC development, we next asked whether Rab5c overactivation could enhance HSPC production. We generated constructs for expressing CA form of zebrafish Rab5c by a Q81L amino acid mutation in the conserved GTP hydrolysis region (Fig 6A), which affected GTPase activity and led to excessive endocytic trafficking [45–47], and its effect of endocytic trafficking overactivation was confirmed by internalization assay in Hela cell line (S6A and S6B Fig). Endosomes induced by GFP-Rab5c CA were larger and more active (S7 Movie) compared with that in GFP-Rab5c WT group (S5 Movie), further confirming the properties. Surprisingly, we found that mRNA injection of *rab5c* CA form but not *rab5c* WT form led to decreased expression of *runx1* (Fig 6B and 6C, S6C Fig). Similarly, the number of $kdrl^+cmyb^+$ definitive hematopoietic precursors within the VDA region was decreased in Rab5c CA group (Fig 6D and 6E). Rab5c CA overexpression driven by *fli1a* promoter also resulted in reduced HSPC production (Fig 6F and 6G), suggesting that the effect of Rab5c CA on HSPC development is EC autonomous. Furthermore, we performed Rab5c CA overexpression by using *hsp70* promoter. WISH showed that *cmyb* was markedly decreased by Rab5c CA HS overexpression at 20 hpf (Fig 6H and 6I), indicating that Rab5c CA impairs HSPC development at the critical phase of HE specification. Taken together, we concluded that Rab5c CA impairs HSPC production.

## Excessive endocytic trafficking induced by Rab5 overactivation impairs Notch signaling during HSPC specification

Previous studies showed that constitutive internalization of LIN-12/Notch by endocytic trafficking attenuated the LIN-12/Notch signaling in Caenorhabditis elegans [72], and suppression of Notch signaling by Numb is mediated through Notch endocytosis [73,74]. We wondered whether Notch signaling was also involved in the HSPC defect induced by *rab5c* CA. WB results showed that the protein level of NICD was decreased in *rab5c* CA overexpression embryos at 26 hpf (Fig 7A and 7B). Live imaging showed that the number of $fli1a^+tp1^+$ Notch-active ECs in the VDA region was markedly reduced in Rab5c CA group (Fig 7C and 7D), consistent with the decreased expression of Notch downstream genes, including *ephrinB2a*, *gata2b*, *hey1*, and *hey2* (Fig 7E–7H). Furthermore, the HSPC defect in Rab5c CA group could be partially rescued by NICD overexpression (Fig 7I and 7J), indicating that Notch signaling inhibition was responsible for HSPC defect in Rab5c CA group. IF in 293T cells showed that the co-localization of JAG1, DLL4, and NOTCH1 with EE marker EEA1 was largely increased in Rab5c CA overexpression cells (Fig 7K–7P), suggesting that Rab5c CA led to excessive endocytic trafficking of Notch ligand and receptor. Moreover, the co-localization of JAG1, DLL4, and Notch1 with LAMP1 in Rab5c CA group was also markedly increased (S6D–S6I Fig). The protein level of Notch ligands and the receptor were obviously decreased in the Rab5c CA group examined by WB (S6J and S6K Fig). Taken together, these results suggested that enhanced endolysosomal trafficking in Rab5c CA group likely impaired Notch signaling by reducing ligands and receptor molecule number.

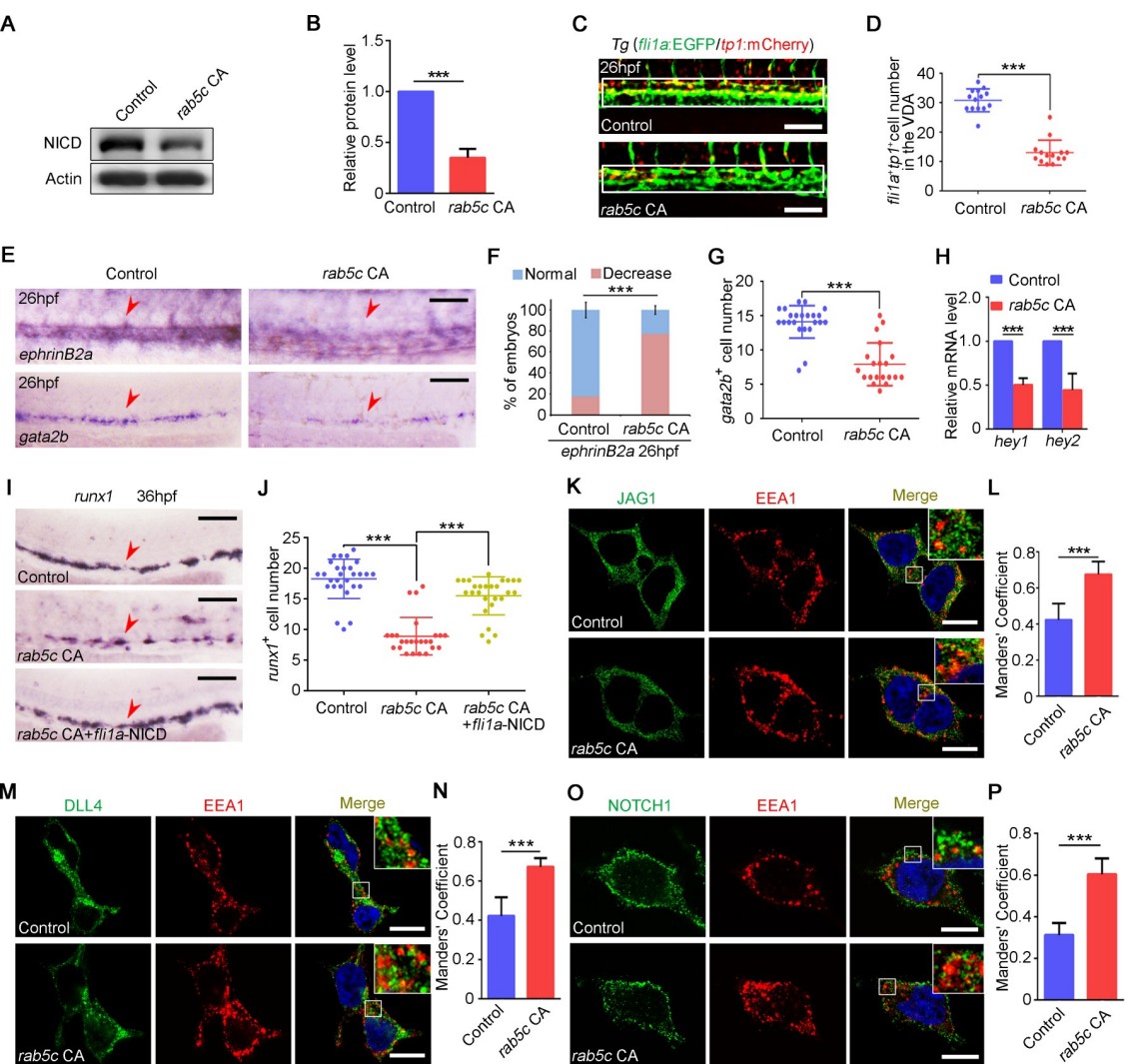

**Fig 7. Excessive endocytic trafficking mediated by Rab5c overactivation impairs Notch signaling during HSPC specification.** (A) Protein level of NICD in control and *rab5c* CA group at 26 hpf examined by WB. (B) Quantification of protein level using gray analysis. Error bars, mean ± SD, ***$P < 0.001$. (C) Confocal imaging shows the *fli1a*$^+$*tp1*$^+$ Notch-active ECs in the VDA region (box area) of control and *rab5c* CA group. Scale bar, 100 μm. (D) Quantification of *fli1a*$^+$*tp1*$^+$ Notch-active ECs. Error bars, mean ± SD, ***$P < 0.001$. (E) Expression of Notch downstream genes *ephrinB2a* and *gata2b* in control and *rab5c* CA group at 26 hpf examined by WISH. Scale bar, 100 μm. (F) Statistical analysis of the WISH. Error bars, mean ± SD, ***$P < 0.001$. (G) Quantification of the *gata2b* positive cells. Error bars, mean ± SD, ***$P < 0.001$. (H) Relative mRNA level of Notch signaling downstream genes *hey1*, *hey2* in the control and the *rab5c* CA group at 26 hpf examined by qRT-PCR. Error bars, mean ± SD, ***$P < 0.001$. (I) WISH analysis shows that *runx1* expression in Rab5c CA HS overexpression group is partially rescued by NICD overexpression through *fli1a*-NICD. Scale bar, 100 μm. (J) Quantification of the *runx1* positive cells. Error bars, mean ± SD, ***$P < 0.001$. (K) Control plasmid or pCS2-*rab5c* CA transfected 293T cells were immunostained with antibodies against endogenous JAG1 (green) and EEA1 (red). Scale bar, 10 μm. (L) Quantification of co-localization of JAG1 with EEA1 using Manders' coefficient (ImageJ). $n = 13$ cells. Error bars, mean ± SD, ***$P < 0.001$. (M) Control plasmid or pCS2-*rab5c* CA transfected 293T cells were immunostained with antibodies against endogenous DLL4 (green) and EEA1 (red). Scale bar, 10 μm. (N) Quantification of co-localization of DLL4 with EEA1 using Manders' coefficient. $n = 12$ cells. Error bars, mean ± SD, ***$P < 0.001$. (O) Control plasmid or pCS2-*rab5c* CA transfected 293T cells were immunostained with antibodies against endogenous NOTCH1 (green) and EEA1 (red). Scale bar, 10 μm. (P) Quantification of co-localization of NOTCH1 with EEA1 using Manders' coefficient. $n = 14$ cells. Error bars, mean ± SD, ***$P < 0.001$. The $P$ values in this figure were calculated by Student *t* test. The underlying data in this figure can be found in S1 Data. CA, constitutively active; EC, endothelial cell; hpf, hours post fertilization; HS, heat shock; HSPC, hematopoietic stem and progenitor cell; NICD, Notch intracellular domain; qRT-PCR, quantitative reverse-transcription PCR; VDA, ventral wall of the dorsal aorta; WB, western blot; WISH, whole-mount in situ hybridization.

## Rab5c overactivation enhances AKT signaling but does not affect EC survival

To determine whether Rab5c CA also leads to AKT signaling alteration as Rab5c DN does, we examined the protein level of total Akt and p-Akt in control and *rab5c* CA group by WB. The results showed that total Akt was not changed; however, the p-Akt was increased in *rab5c* CA group (S7A and S7B Fig). Rab5c CA overexpression driven by *hsp70* promoter showed similar results (S7C and S7D Fig). These results suggested that Rab5c overactivation could enhance AKT signaling. However, TUNEL assay showed that the number of apoptotic *fli1a*+ cells (including ECs and HE cells) in the VDA region was not changed in Rab5c CA group (S7E and S7F Fig). Taken together, we concluded that Rab5c overactivation did not alter EC survival in the VDA region.

## Discussion

Previous studies on Rab5 isoforms–regulated endocytic trafficking mainly utilized in vitro cell cultures, which are convenient for observing intracellular molecular dynamics. However, these studies are not able to examine the developmental or physiological effect of endocytic trafficking in vivo. In this study, we use zebrafish, the in vivo model, to explore the role of Rab5c in HSPC development and have uncovered the function of endocytic trafficking in this process. We have shown that the endocytic trafficking regulator Rab5c is highly expressed in the VDA region during the critical period of HE specification and is required for HSPC production. The deficiency or overactivation of Rab5c leads to impaired Notch signaling, further resulting in HE specification defect. Moreover, Rab5c-deficiency–induced dysfunction of AKT signaling disrupts HE survival. Collectively, these findings suggest that Rab5c-mediated endocytic trafficking may not only play an instructive role in HE specification but also serve as a permissive regulator for HE survival.

In vertebrates, cell autonomous Notch signaling is indispensable for ECs transitioning to HE cells, which further give rise to HSPCs [4,18,26]. Several regulators for Notch in HE specification have been identified [55,75]. However, it still remains elusive how the precise regulations are achieved. Endocytic trafficking changes the spatial distribution and number of plasma membrane-bound signal molecules at a specific time through plasma membrane to endosome transport, interendosome transport, and endosome to lysosome transport, which is crucial for spatiotemporal regulation of signal transduction. Furthermore, the endosome not only acts as specialized zone communicating with the plasma membrane and lysosome but also provides appropriate environment for biochemical reaction [33,76]. Previous studies provide some clues about Notch regulation mediated by endocytosis [30,32]. However, how Notch signaling is spatial-temporally regulated in HSPC development still remains elusive. In this work, we demonstrated that Rab5-mediated endocytic trafficking regulates Notch signaling to ensure suitable endosome environment for subsequent NICD release in HSPC production.

A recent study showed that RAB5C regulated trafficking of the CD93/Multimerin/β1-integrin complex in human umbilical vein ECs (HUVECs) and human dermal blood ECs (HDBECs) [77]. Furthermore, another study demonstrated that Rap1b promoted Notch-mediated HE specification by enhancing integrin-mediated cell adhesion during somitic wave of HSPC specification [78]. In addition, previous studies showed that HS induction of Notch after 16 hpf can not rescue HSPC defects induced by repression of early somitic Notch signaling [50,56]. In our study, induction of Notch signaling by HS-induced overexpression of NICD at 20 hpf can rescue HSPC defects in Rab5c-deficiency embryos, suggesting that Rab5c regulates Notch during the late endothelial wave (after 20 hpf) but not early somitic wave (15–

17 hpf). Furthermore, analysis of *rab5c* expression pattern showed that *rab5c* was not specifically expressed at 14 hpf or 16 hpf but was specifically expressed in ECs at 20 hpf or 26 hpf. Taken together, we propose that Rab5c may not be involved in integrin-mediated somitic wave regulation.

Rab5c DN and CA constructs have opposite activities for Rab5; however, their effects are similar in terms of Notch activation during HSPC development. Rab5c DN overexpression resulted in endocytic trafficking inhibition and reduced EE localization of Notch ligand and receptor. In normal circumstances, EE localization of Notch is a required step for activation of Notch [30,32]. Therefore, the impaired endocytic trafficking of Notch ligand and receptor may be the reason for Notch signaling defect in Rab5c DN group. However, Rab5c CA likely leads to a special circumstance in which largely increased EE localization of Notch enhances endolysosomal trafficking, further leading to a dramatic reduction of Notch ligand and receptor molecules. Our results support the view that excessive endocytic trafficking of Notch attenuates the Notch signaling [72]. Therefore, we speculate that the balanced endocytic trafficking of Notch mediated by Rab5c is important for Notch activation and HSPC specification in zebrafish.

It has been known that Rab5*c* can also regulate another type of endosome labeled by Appl1, which is different from endosomes labeled by EEA1. Our data show that AKT signaling activation mediated by the complex of Rab5c and Appl1 is required for HE survival. These results suggest that Rab5c may simultaneously regulate Notch and AKT signaling by 2 types of endosomes, and the coordinated regulation of these signaling pathways is crucial for HSPC production. The cell autonomous Notch signaling regulated by Rab5c may play an instructive role in HE specification, whereas the AKT signaling seems to provide a permissive signal to maintain HE survival.

In our RNA-seq data, multiple signaling pathways were down-regulated and up-regulated in Rab5c-deficiency ECs. Among these signaling pathways, Notch and AKT signaling are crucial for HSPC development [18,26,34,35]. However, other altered signals, such as oxidative phosphorylation, Arp2/3 protein complex, lipoprotein particle, NF-kappa B signaling, and tight junction, also exist in the RNA-seq data. These signals are involved in physiological-biological activities of energy metabolism, cytoskeleton remodeling, lipid metabolism, inflammation, and cell-to-cell contact, which may also be involved in the HSPC defect induced by Rab5c-deficiency. Whether these signals are regulated by Rab5c as well as play essential roles in HSPC development awaits further investigation.

Taken together, our study, for the first time, shows that Rab5c is crucial for HSPC development through maintaining appropriate endocytic trafficking state, i.e., endocytic trafficome, during early embryogenesis. The appropriate endocytic trafficome may simultaneously regulate Notch and AKT in the VDA region during the critical phase of HE specification, leading to these signals being well orchestrated. These findings may provide some insights into how ECs process the spatiotemporally and elaborately orchestrated signals and accurately execute fate transition to HSPCs.

## Materials and methods

### Ethics statement

Our zebrafish experiments were all approved and carried out in accordance with the Animal Care Committee at the Institute of Zoology, Chinese Academy of Sciences (permission number: IOZ-13048).

## Zebrafish husbandry

Zebrafish strains, including AB, Tubingen, *Tg*(*kdrl*:mCherry) [5], *Tg*(*fli1a*:EGFP) [79], *Tg* (*runx1*:en-GFP) [39], *Tg*(*cmyb*:GFP) [80], *Tg*(*CD41*:GFP) [81], *Tg*(*tp1*:mCherry) [82], *Tg*(*gfi1*: GFP) [83], *Tg*(TOPflash:GFP) [84], were raised in system water at 28.5˚C under standard conditions. The zebrafish embryos were acquired by natural spawning.

## MOs, vector construction, and mRNA overexpression

MOs used in this study were purchased from GeneTools (Philomath, Oregon). The MO sequences: *rab5c* MO: 5′-CATGCCAACAGGCTGGACAACAGGA-3′; *rab5ab* MO: 5′-TCGTTGCTCCACCTCTTCCTGCCAT-3′ [85]; *rab5b* MO: 5′-CCTGCCTGTCCCACGGG TACTCATG-3′ [85]; *tp53* MO: 5′-GCGCCATTGCTTTGCAAGAATTG-3′ [86]; *appl1* MO: 5′-TAGTTTATCGATTCCAGGCATGGCT-3′ [38]; Control MO: 5′-CCTCTTACCTCAGTT ACAATTTATA-3′ [87]. MOs were injected into one-cell stage embryos. For mRNA overexpression, Flag-tagged *rab5c* and *AKT2* CA (myristylation signal MGSSKSKPK at its N terminal) CDS were cloned into pCS2$^+$ vector, *rab5c* DN (S36N) and CA (Q81L) were generated by PCR-mediated point mutation, mRNAs were generated using SP6 mMessage Machine kit (AM1340; Ambion, Austin, Texas) and injected into one-cell stage embryos. The primers for point mutation are listed in S1 Table. For endothelium specific overexpression of GFP-*rab5c* DN, CA and WT (lacking *rab5c* MO binding site), GFP-*rab5c* DN, CA and WT were cloned into pDONR221 vector by BP reaction (Gateway BP Clonase II Enzyme mix, 11789100; Invitrogen, Carlsbad, California), then subcloned into pDestTol2pA2 vector [88] with *fli1a* promoter [89] by LR reaction (LR Clonase II Plus enzyme, 12538200; Invitrogen). For somite-specific overexpression of *rab5c* mRNA lacking the MO binding site, GFP-*rab5c* WT (lacking *rab5c* MO binding site) was cloned into pDestTol2pA2 vector with *mylz* promoter by NEBuilder® HiFi DNA Assembly Master Mix (E2621S; NEB, Ipswich, Massachusetts). For HS induction of GFP-Rab5c DN and CA, GFP-*rab5c* DN and CA were cloned into pHSP vector with an *hsp70* promoter. For highly efficient overexpression, the plasmids were co-injected with Tol2 mRNA into one-cell stage embryos as previously described [88]. The HS condition was 30 min at 42˚C.

## WISH and double FISH

WISH was carried out using a ZF-A4 in situ hybridization machine (Zfand, Beijing, China) with DIG (11277073910; Roche, Basel, Switzerland) labeled single-stranded RNA probe. For probe synthesis, gene-specific PCR products were cloned into pGEM-T vector (A3600; Promega, Madison, Wisconsin), RNA probe was transcribed with T7 RNA polymerase (P2075; Promega). After hybridization, the RNA was recognized by AP-conjugated anti-DIG antibody (11093274910; Roche), and color reaction was carried out using BM purple (11442074001; Roche) as the substrate. For double FISH assay, one probe was labeled with DIG and detected by POD-conjugated anti-DIG antibody and the other probe was labeled with fluorescein and detected by POD-conjugated anti-fluorescein antibody. The TSA Plus Cy3 Solution (NEL744001KT; PerkinElmer, Waltham, Massachusetts) and TSA Plus Fluorescein Solution (NEL741001KT; PerkinElmer) were used as the substrate, respectively.

## Generation of mutants by CRISPR/Cas9

The *rab5c*, *rab5ab*, and *rab5b* mutants were generated using CRISPR/Cas9; the method for Cas9 mRNA and guide RNA synthesis was described previously [90]. pXT7-Cas9 was used for Cas9 mRNA transcription; capped Cas9 mRNA was generated using T7 mMessage Machine

kit (AM1344; Ambion). Then Cas9 mRNA was purified using RNA clean Kit (DP412; TIAN-GEN, Beijing, China), and gRNA was generated using in vitro transcription by T7 RNA polymerase (P2075; Promega). The gRNA target sequences are listed in S2 Table. Mutant identification was carried out by high-resolution melting curve (HRM) and DNA sequencing analysis. The primers used for mutant screening are listed in S3 Table.

## qRT-PCR

Relative abundance of target mRNAs was examined by qRT-PCR. A total of 2 μg RNA extracted from zebrafish embryonic trunk region tissue was used for cDNA synthesis using M-MLV Reverse Transcriptase (M1705; Promega). Experiments were performed according to the qPCR manufacturer's instructions (FP205-03; TIANGEN) and examined by a CFX96 Real-Time PCR System (Bio-Rad, Orange, California). The primers used are summarized in S3 Table.

## Cell culture and transfection

Human HEK-293T and Hela cell lines were maintained in DMEM (SH30022; Thermo, Waltham, Massachusetts), supplemented with 10% FBS (10099141; Thermo), 100 U/mL penicillin, and 100 mg/mL streptomycin (SV30010; Thermo) at 37˚C, 5% $CO_2$. The plasmids were transfected into cells using lipofectamine 3000 (L3000015; Invitrogen), according to the manufacturer's instructions.

## TRITC-transferrin internalization assay

Cells transfected with indicated plasmids were serum-starved for 6 h, then kept on ice for 10 min, followed by incubation in PBS containing 10 μg/mL of TRITC-conjugated transferrin (009–0034; Rockland, Philadelphia, Pennsylvania) and 1% BSA (B2064; Sigma-Aldrich, Saint Louis, Missouri) for 10 min at 37˚C for transferrin internalization, then washed 3 times with cold PBS. Nuclei were visualized by staining with Hoechst 33342 (H3570; Invitrogen). Pictures were taken by Nikon A1 confocal microscopy (Nikon, Tokyo, Japan).

## Microscopy imaging

Fluorescent images were taken with a Nikon A1 confocal microscope (Nikon). Microscopic observation and photography were carried out as previously described [91]. The relative mean fluorescence intensities and Manders' coefficients were analyzed using ImageJ.

## RNA-seq and GO analysis

The trunk region tissue from $Tg(fli1a$:EGFP) background control and $rab5c$ morphants was dissected and dissociated into single cells. $fli1a^+$ cells were sorted using MoFlo XDP (BECK-MAN, Brea, California). RNA extracted using RNeasy Mini kit (74104; QIAGEN, Hilden, German) was reversely transcribed. cDNA samples were sequenced using Illumina NovaSeq 6000. The quality of raw sequencing reads was screened using FastQC. Quality of raw RNA-seq reads for each sample by FastQC software, and low-quality bases were trimmed and filtered by cutadapt and Trimmomatic. The remaining reads were mapped to the *Danio rerio* gene information downloaded from the National Center for Biotechnology Information database. The fold change of expression level of transcripts was calculated using DEGseq package. GO analysis was performed using the DAVID website (https://david.ncifcrf.gov/). The original RNA-seq data of ECs ($kdrl^+runx1^-$) and HE cells ($kdrl^+runx1^+$) are from Zhang and colleagues [39].

## TUNEL assay

The TUNEL assay was conducted as previously described [92] with some modifications. In brief, the control or *rab5c* MO injected *Tg*(*fli1a*:EGFP) transgenic embryos were fixed in 4% PFA (MK104005; Merck, Darmstadt, Germany) and then dehydrated with methanol at −20˚C overnight. After rehydration, the embryos were washed with PBST 3 times and treated with 10 μg/ml Proteinase K (0706–100; Amersco, Solon, Ohio) for permeabilization. The permeabilized embryos were fixed in 4% PFA for 30 min at room temperature. After washing 3 times with PBST, the embryos were incubated with a TUNEL labeling mixture (In Situ Cell Death Detection Kit TMR Red, 11767305001; Roche) at 4˚C overnight. After washing with PBST, the embryos were photographed by confocal microscopy.

## Co-IP assays

Co-IP assays were performed as previously described [39] with some modifications. Briefly, transfected HEK-293T cells were harvested and lysed in IP lysis buffer (50 mM Tris-HCl [pH 7.4], 150 mM NaCl, 1 mM EDTA, 0.5 mM DTT, 0.5% NP-40, 5%[v/v] Glycerol, Roche cocktail protease inhibitor). The lysate was incubated with anti-FLAG M2 Affinity Gel (A2220, Sigma-Aldrich) for 4 hours at 4˚C. After washing, the samples were eluted with elution buffer and examined by WB with anti-Flag (F7425; Sigma-Aldrich) and anti-Myc (06–549; Millipore, Billerica, Massachusetts) antibodies.

## WB and immunofluorescence

WB experiments were performed as previously reported [69,93] using the following antibodies: anti-β-Actin antibody (4967; Cell Signaling Technology, Danvers, Massachusetts), anti-NICD antibody (ab83232; Abcam, Cambridge, UK), anti-Myc antibody (06–549; Millipore), anti-Flag antibody (F7425; Sigma-Aldrich), anti-DLL4 antibody (ab7280; Abcam), anti-JAG1 antibody (ab7771; Abcam), anti-Notch1 antibody (3608; Cell Signaling Technology), anti-Erk antibody (9102; Cell Signaling Technology), anti-p-Erk antibody (9101; Cell Signaling Technology), anti-p-AKT antibody (4060; Cell Signaling Technology), and anti-AKT antibody (9272; Cell Signaling Technology). Quantification of protein level using gray analysis (Gel-Pro analyzer). Immunofluorescence experiments were performed as previously reported [94] using the following antibodies: anti-Notch1 antibody (3608; Cell Signaling Technology), anti-Notch1 antibody (3447; Cell Signaling Technology), anti-DLL4 antibody (ab7280; Abcam), anti-JAG1 antibody (ab7771; Abcam), anti-LAMP1 antibody (15665; Cell Signaling Technology), anti-EEA1 antibody (610457; BD Biosciences, Franklin Lakes, New Jersey), anti-APPL1 antibody (3858; Cell Signaling Technology), anti-AKT antibody (2920; Cell Signaling Technology), anti-AKT antibody (9272; Cell Signaling Technology), and anti-PIK3CA antibody (ab40776; Abcam).

## Statistical analysis

The Student *t* test was used for statistical comparisons. Sample sizes were indicated in the figures or figure legends. Plotted mean was calculated by GraphPad software. Data were shown as mean ± SD. *P* values were used for significance evaluation. $^*P < 0.05$, $^{**}P < 0.01$, $^{***}P < 0.001$.

## Supporting information

**S1 Fig. Expression of Rab5c is enriched in ECs of trunk region.** (A) Flowchart of FACS and qRT-PCR analysis of endothelial and nonendothelial cells in the trunk region from *Tg*(*fli1a*:

EGFP) transgenic zebrafish embryos at 26 hpf. The trunk region in this transgenic line was dissected and dispersed into single cells for FACS and qRT-PCR. (B) qRT-PCR analysis of *fli1a⁻* and *fli1a⁺* cells. The *flk1* was used as positive control for endothelium specific expression gene. The expression of *rab5c* is enriched in *fli1a⁺* ECs. *P* value was calculated by Student *t* test, **$P < 0.01$,***$P < 0.001$. The underlying data in this figure can be found in S1 Data. EC, endothelial cell; FACS, fluorescence-activated cell sorting; hpf, hours post fertilization; qPT-PCR, quantitative reverse-transcription PCR
(TIF)

**S2 Fig. Rab5c is required for definitive hematopoiesis.** (A) Design of *rab5c* MO targeting the intersection of 5′UTR and CDS of *rab5c* mRNA for translation blocking. The start codon of *rab5c* is in blue. (B) KD efficiency of *rab5c* MO examined by WB. Flag-tagged mRNA containing the *rab5c* MO binding site and full-length *rab5c* CDS was co-injected with either control or *rab5c* MO into one-cell stage embryos. Rab5c-Flag was detected by anti-Flag antibody. (C) Quantification of protein level using gray analysis (Gel-Pro analyzer). Error bars, mean ± SD, ***$P < 0.001$. (D) Rab5c-deficiency does not impair primitive hematopoiesis. Expression of *scl* in hemangioblast, *gata1* in red blood cells, and *pu.1* in myeloid cells is not changed in *rab5c* morphants compared with control. This examination was carried out using WISH. The numbers below the WISH pictures mean number of embryos showing representative phenotype/ total number of embryos. Scale bar, 100 μm. (E) Flag-tagged *rab5c* mRNA lacking the *rab5c* MO binding site was co-injected with either control or *rab5c* MO into one-cell stage embryos. The protein level was examined by WB. (F) Quantification of protein level using gray analysis (Gel-Pro analyzer). Error bars, mean ± SD. (G) HSPC rescue of *rab5c* morphants with *rab5c* mRNA. *rab5c* mRNA lacking the *rab5c* MO binding site can rescue the expression of HSPC marker *runx1* in *rab5c* morphants. The red arrowheads denote HSPCs. Scale bar, 100 μm. (H) Snapshot in S4 Movie. Time-lapse imaging shows EHT process in *rab5c* morphants. The arrow denotes the cell undergoing EHT progress. Scale bar, 100 μm. (I) Relative mRNA level of other zebrafish Rab5 family genes *rab5aa*, *rab5ab*, *rab5b* in *rab5c* WT, mutant, and *rab5c* morphants at 26 hpf examined by qRT-PCR. Error bars, mean ± SD, *$P < 0.05$. (J) WISH results show that expression of *runx1* is unchanged in low-dose of *rab5ab* and *rab5b* MOs co-injected *rab5c* WT embryos but is severely decreased in low-dose of MOs co-injected *rab5c* mutant embryos. Scale bar, 100 μm. (K) Generation of *rab5ab* mutant using the CRISPR/Cas9 technique. *rab5ab* WT and mutant sequences are listed. (L) Generation of *rab5b* mutant using the CRISPR/Cas9 technique. *rab5b* WT and mutant sequences are listed. (M) Expression of *runx1* is not changed in *rab5ab* mutant embryos compared with WT sibling. Scale bar, 100 μm. (N) Expression of *runx1* is not changed in *rab5b* mutant embryos compared with WT sibling. Scale bar, 100 μm. (O) Relative mRNA level of Rab5 family genes *rab5aa*, *rab5b*, and *rab5c* in *rab5ab* WT or mutant embryos at 26 hpf examined by qRT-PCR. Error bars, mean ± SD. (P) Relative mRNA level of Rab5 family genes *rab5aa*, *rab5ab*, and *rab5c* in *rab5b* WT or mutant embryos at 26 hpf examined by qRT-PCR. Error bars, mean ± SD. (Q) Expression of *runx1* in WT sibling and *rab5ab*/*rab5c* double-knockout embryos examined by WISH. HSPC specification is severely impaired in *rab5ab*/*rab5c* double-knockout embryos. Scale bar, 100 μm. (R) Expression of *runx1* in WT sibling and *rab5b*/*rab5c* double-knockout embryos examined by WISH. HSPC specification is severely impaired in *rab5b*/*rab5c* double-knockout embryos. Scale bar, 100 μm. The *P* values in this figure were calculated by Student *t* test. The underlying data in this figure can be found in S1 Data. CDS, coding sequence; EHT, endothelial-to-hematopoietic transition; hpf, hours post fertilization; HSPC, hematopoietic stem and progenitor cell; KD, knockdown; MO, morpholino; n.s., nonsignificant; qRT-PCR, quantitative reverse-transcription PCR; WB, western blot; WISH, whole-mount in situ hybridization;

WT, wild type
(TIF)

**S3 Fig. Rab5c function is in an EC autonomous manner.** (A) TRITC-conjugated TF internalization assay in Hela cells transfected with empty pCS2 or pCS2-*rab5c* DN plasmids. Representative pictures were shown. Scale bar, 10 μm. (B) Quantitative fluorescence intensity of intracellular TRITC-TF in empty pCS2 or pCS2-*rab5c* DN transfected Hela cells, *n* = 8 cells for each group. Error bars, mean ± SD. *P* value was calculated by Student *t* test, ***$P < 0.001$. (C) Fluorescence microscope imaging shows that the GFP expression is detected by 2 hours post HS at 20 hpf in *hsp70*-GFP-*rab5c* DN group, but not in control. Scale bar, 200 μm. (D) Fluorescence microscope imaging shows that the GFP expression is detected in ECs of *fli1a*-GFP-*rab5c* DN group, but not in control. Scale bar, 200 μm. (E) Fluorescence microscope imaging shows that the GFP expression is detected in somitic cells of *mylz*-GFP-*rab5c* group but not in the control. Scale bar, 200 μm. (F) Expression of *dlc*, *vegfa*, and *shh* in control and *rab5c* morphants examined by WISH. Scale bars, 400 μm. (G) Statistical analysis of the WISH. Error bars, mean ± SD. The *P* values in this figure were calculated by Student *t* test. The underlying data in this figure can be found in S1 Data. DN, dominant-negative; EC, endothelial cell; GFP, green fluorescent protein; hpf, hours post fertilization; HS, heat shock; n.s., nonsignificant; TF, transferrin; TRITC, tetramethylrhodamine; WISH, whole-mount in situ hybridization
(TIF)

**S4 Fig. Rab5c regulates HSPC development through endothelial Notch signaling.** (A) Expression of panvascular gene *flk1*, Notch ligand gene *dll4*, and Notch-independent arterial marker *tbx20* in the control and the *rab5c* morphants at 26 hpf examined by WISH. The numbers below the WISH pictures mean number of embryos showing representative phenotype/total number of embryos. Scale bar, 100 μm. (B) qRT-PCR results show that NICD overexpression leads to Notch downstream gene *hey1* and *hey2* up-regulation. NICD overexpression was carried out by *hsp70*-NICD HS at 20 hpf. (C) WISH results show that NICD overexpression leads to up-regulation of *ephrinB2a* positive ECs and *cmyb*-marked HSPCs. Scale bar, 100 μm. (D) WISH shows that *runx1* expression in *rab5c* morphants is partially rescued by NICD overexpression through *hsp70*-NICD-2a-GFP HS at 20 hpf. Scale bar, 100 μm. (E) Fluorescence microscope imaging shows the GFP expression. Scale bar, 400 μm. (F) Expression of *runx1* in control and *rab5c* morphants examined by WISH. Scale bars, 100 μm. (G) Quantification of the *runx1*+ cells. Error bars, mean ± SD, ***$P < 0.001$. (H) The expression pattern of *rab5c* during zebrafish early developmental stage examined by WISH. Scale bars, 400 μm. (I) Control empty plasmid or pCS2-*rab5c* DN transfected 293T cells were immunostained with antibodies against endogenous JAG1 (green) and LAMP1 (red). Scale bar, 10 μm. (J) Quantification of co-localization of JAG1 with LAMP1 using Manders' coefficient (ImageJ). *n* = 14 cells. Error bars, mean ± SD, ***$P < 0.001$. (K) Control plasmid or pCS2-*rab5c* DN transfected 293T cells were immunostained with antibodies against endogenous DLL4 (green) and LAMP1 (red). Scale bar, 10 μm. (L) Quantification of co-localization of DLL4 with LAMP1 using Manders' coefficient. *n* = 14 cells. Error bars, mean ± SD, **$P < 0.01$. (M) Control plasmid or pCS2-*rab5c* DN transfected 293T cells were immunostained with antibodies against endogenous NOTCH1 (green) and LAMP1 (red). Scale bar, 10 μm. (N) Quantification of co-localization of NOTCH1 with LAMP1 using Manders' coefficient. *n* = 14 cells. Error bars, mean ± SD, ***$P < 0.001$. (O) Protein level of NOTCH1, DLL4, and JAG1 in control empty pCS2 or pCS2-*rab5c* DN transfected 293T cells examined by WB. (P) Quantification of protein level using gray analysis (Gel-Pro analyzer). Error bars, mean ± SD. The *P* values in this figure were calculated by Student *t* test. The underlying data in this figure can be found in S1 Data. DN, dominant-negative; EC, endothelial cell; GFP, green fluorescent protein; hpf, hours post

fertilization; HS, heat shock; HSPC, hematopoietic stem and progenitor cell; NICD, Notch intracellular domain; n.s., nonsignificant; qRT-PCR, quantitative reverse-transcription PCR; WB, western blot; WISH, whole-mount in situ hybridization
(TIF)

**S5 Fig. Rab5c regulates AKT but not Wnt, Erk, Shh, or Vegf pathway.** (A) Protein level of p-Akt and total Akt in control and Rab5c inhibition group at 26 hpf examined by WB. *rab5c* DN overexpression was carried out by *hsp70*-GFP-*rab5c* DN HS at 14 or 20 hpf. (B) Quantification of protein level using gray analysis (Gel-Pro analyzer). Error bars, mean ± SD, $^{**}P < 0.01$, $^{***}P < 0.001$. (C) 293T cells were immunostained with antibodies against endogenous AKT (green), APPL1 (red), or EEA1 (red). The nucleus was stained with Hoechst. Scale bar, 10 μm. (D) Quantification of co-localization of target proteins using Manders' coefficient (ImageJ). $n = 14$ cells. Error bars, mean ± SD, $^{***}P < 0.001$. (E) 293T cells were immunostained with antibodies against endogenous NOTCH1 (green), EEA1 (red), or APPL1 (red). The nucleus was stained with Hoechst. Scale bar, 10 μm. (F) Quantification of co-localization of target proteins using Manders' coefficient (ImageJ). $n = 14$ cells. Error bars, mean ± SD, $^{***}P < 0.001$. (G) Expression of *runx1* in control and *appl1* morphants examined by WISH. Scale bar, 100 μm. The numbers below the WISH pictures mean: number of embryos showing representative phenotype/total number of embryos. (H) Examination of Wnt activity by *Tg* (TOPflash:GFP) line. Scale bar, 100 μm. (I) Examination of Wnt signaling downstream gene *c-myc*, *mycbp*, and *cldn1* expression by qRT-PCR. Error bars, mean ± SD. (J) Protein level of Erk and p-Erk in control and *rab5c* morphants at 26 hpf. (K) Quantification of protein level using gray analysis (Gel-Pro analyzer). Error bars, mean ± SD. (L) Expression of *shh* and *vegfa* in control and *rab5c* morphants examined by WISH. Scale bars, 100 μm. (M) Statistical analysis of the WISH examining *shh*. Error bars, mean ± SD. (N) Statistical analysis of the WISH examining *vegfa*. Error bars, mean ± SD. The *P* values in this figure were calculated by Student *t* test. The underlying data in this figure can be found in S1 Data. DN, dominant-negative; hpf, hours post fertilization; HS, heat shock; n.s., nonsignificant; p-Akt, phosphorylated Akt; p-Erk, phosphorylated Erk; qRT-PCR, quantitative reverse-transcription PCR; WB, western blot; WISH, whole-mount in situ hybridization
(TIF)

**S6 Fig. Rab5c over activation leads to excessive endolysosomal trafficking of Notch ligands and receptor.** (A) TRITC-conjugated TF internalization assay in Hela cells transfected with control empty pCS2 or pCS2-*rab5c* CA plasmid. Representative pictures were shown. Scale bar, 10 μm. (B) Quantitative fluorescence intensity of intracellular TRITC-TF in control empty pCS2 and pCS2-*rab5c* CA transfected cells, $n = 8$ cells for each group. Error bars, mean ± SD. *P* value was calculated by Student *t* test, $^{***}P < 0.001$. (C) *rab5c* WT mRNA overexpression does not lead to HSPC production alteration. WISH results show that the expression of *runx1* is not changed in *rab5c* WT mRNA overexpression group compared with control. The numbers below the WISH pictures mean: number of embryos showing representative phenotype/total number of embryos. Scale bar, 100 μm. (D) Control empty pCS2 or pCS2-*rab5c* CA transfected 293T cells were immunostained with antibodies against endogenous JAG1 (green) and LAMP1 (red). Scale bar, 10 μm. (E) Quantification of co-localization of JAG1 with LAMP1 using Manders' coefficient (ImageJ). $n = 14$ cells. Error bars, mean ± SD, $^{***}P < 0.001$. (F) Control plasmid or pCS2-*rab5c* CA transfected 293T cells were immunostained with antibodies against endogenous DLL4 (green) and LAMP1 (red). Scale bar, 10 μm. (G) Quantification of co-localization of DLL4 with LAMP1 using Manders' coefficient. $n = 14$ cells. Error bars, mean ± SD, $^{*}P < 0.05$. (H) Control plasmid or pCS2-*rab5c* CA transfected 293T cells were immunostained with antibodies against endogenous NOTCH1 (green) and LAMP1 (red).

Scale bar, 10 μm. (I) Quantification of co-localization of NOTCH1 with LAMP1 using Manders' coefficient. $n$ = 14 cells. Error bars, mean ± SD, ***$P$ < 0.001. (J) Protein level of NOTCH1, DLL4, JAG1 in control empty pCS2 or pCS2-*rab5c* CA transfected 293T cells examined by WB. (K) Quantification of protein level using gray analysis (Gel-Pro analyzer). Error bars, mean ± SD, **$P$ < 0.01. The $P$ values in this figure were calculated by Student $t$ test. The underlying data in this figure can be found in S1 Data. CA, constitutively active; HSPC, hematopoietic stem and progenitor cell; TF, transferrin; TRITC, tetramethylrhodamine; WB, western blot; WISH, whole-mount in situ hybridization; WT, wild type
(TIF)

**S7 Fig. Rab5c over-activation leads to AKT signaling up-regulation but does not affect EC survival during HSPC development.** (A) Protein level of p-Akt and total Akt in control and *rab5c* CA mRNA injected embryos examined by WB. (B) Quantification of protein level using gray analysis (Gel-Pro analyzer). Error bars, mean ± SD, **$P$ < 0.01. (C) Protein level of p-Akt and total Akt in control and Rab5c CA group examined by WB. *hsp70-rab5c* CA injected embryos were heat-shocked at 14 hpf or 20 hpf for Rab5c CA overexpression. (D) Quantification of protein level using gray analysis. Error bars, mean ± SD, **$P$ < 0.01. (E) TUNEL assay shows the apoptotic *fli1a*$^+$ (yellow) cells in the VDA region (box area) of control and *rab5c* CA mRNA overexpression group. Scale bar, 100 μm. (F) Quantification of TUNEL$^+$ ECs. Error bars, mean ± SD. The $P$ values in this figure were calculated by Student $t$ test. The underlying data in this figure can be found in S1 Data. CA, constitutively active; EC, endothelial cell; hpf, hours post fertilization; HSPC, hematopoietic stem and progenitor cell; n.s., nonsignificant; p-Akt, phosphorylated Akt; TUNEL, terminal-deoxynucleoitidyl transferase mediated nick end labeling; VDA, ventral wall of the dorsal aorta; WB, western blot
(TIF)

**S1 Raw Images. The uncropped blots for the westerns.**
(PDF)

**S1 Movie. Blood flow in control MO injected embryos.** Time-lapse imaging was performed to observe the blood flow in control morphants. Scale bar, 100 μm. MO, morpholino
(MP4)

**S2 Movie. Blood flow in *rab5c* MO injected embryos.** Time-lapse imaging was performed to observe the blood flow in *rab5c* morphants. Scale bar, 100 μm. MO, morpholino
(MP4)

**S3 Movie. EHT process in control MO injected embryos.** Time-lapse imaging was performed to observe the EHT process in control morphants. Scale bar, 100 μm. EHT, endothelial-to-hematopoietic transition; MO, morpholino
(MP4)

**S4 Movie. EHT process in *rab5c* MO injected embryos.** Time-lapse imaging was performed to observe the EHT process in *rab5c* morphants. The few *kdrl*$^+$*cmyb*$^+$ cells in *rab5c* morphants were able to transform to spherical shape and emerge from the DA. Scale bar, 100 μm. DA, dorsal aorta; EHT, endothelial-to-hematopoietic transition; MO, morpholino
(MP4)

**S5 Movie. Rab5c WT aggregated in a characteristic of endosomal pattern.** Time-lapse imaging was performed to observe the Hela cells transfected with pCS2-GFP-*rab5c* WT plasmid. Nuclei were visualized by staining with Hoechst. The GFP-Rab5c WT aggregated in a

characteristic of endosomal pattern. Scale bar, 10 μm. GFP, green fluorescent protein; WT, wild type
(MP4)

**S6 Movie. Rab5c DN was not able to aggregate in a characteristic of endosomal pattern.**
Time-lapse imaging was performed to observe the Hela cells transfected with pCS2-GFP-*rab5c* DN plasmid. Nuclei were visualized by staining with Hoechst. GFP-Rab5c DN was not able to form the vesicular structures. Scale bar, 10 μm. DN, dominant-negative; GFP, green fluorescent protein
(MP4)

**S7 Movie. Endosomes induced by GFP-Rab5c CA were larger and more active compared with that in GFP-Rab5c WT group.** Time-lapse imaging was performed to observe the Hela cells transfected with pCS2-GFP-*rab5c* CA plasmid. Nuclei were visualized by staining with Hoechst. GFP-Rab5c CA overexpression led to formation of larger and more active endosomes. Scale bar, 10 μm. CA, constitutively active; GFP, green fluorescent protein; WT, wild type
(MP4)

**S1 Table. The primers for point mutation experiments.**
(DOC)

**S2 Table. The gRNA target and PAM sequences.** gRNA, guide RNA; PAM, protospacer adjacent motif
(DOC)

**S3 Table. The primers for qRT-PCR and mutant screening.** qRT-PCR, quantitative reverse-transcription PCR
(DOC)

**S1 Data. Numerical data used in Figs 1, 2, 3, 4, 5, 6 and 7 and S1, S2, S3, S4, S5, S6 and S7 Figs.**
(XLSX)

## Acknowledgments

We thank Dr. Qiang Wang who provided the *Tg*(TOPflash:GFP) transgenic line.

## Author Contributions

**Conceptualization:** Jian Heng, Feng Liu.

**Funding acquisition:** Feng Liu.

**Investigation:** Jian Heng, Peng Lv, Yifan Zhang, Xinjie Cheng, Lu Wang, Dongyuan Ma.

**Project administration:** Feng Liu.

**Supervision:** Feng Liu.

**Writing – original draft:** Jian Heng, Feng Liu.

**Writing – review & editing:** Feng Liu.

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
