## [Editor Report · Decision Letter 0]

12 Nov 2019

Dear Dr Liu, 

Thank you for submitting your manuscript entitled "Rab5c-mediated endocytic trafficking regulates hematopoietic stem and progenitor cell development via Notch and AKT signaling" for consideration as a Research Article by PLOS Biology.

Your manuscript has now been evaluated by the PLOS Biology editorial staff as well as by an academic editor with relevant expertise and I am writing to let you know that we would like to send your submission out for external peer review.

Please re-submit your manuscript within two working days, i.e. by Nov 14 2019 11:59PM.

Kind regards,

Di Jiang

PLOS Biology

---

## [Decision Letter · Decision Letter 1]

9 Dec 2019

Dear Dr Liu,

Thank you very much for submitting your manuscript "Rab5c-mediated endocytic trafficking regulates hematopoietic stem and progenitor cell development via Notch and AKT signaling" for consideration as a Research Article at PLOS Biology. Your manuscript has been evaluated by the PLOS Biology editors, an Academic Editor with relevant expertise, and by three independent reviewers.

In light of the reviews (below), we would welcome re-submission of a much-revised version that takes into account the reviewers' comments. Please address all reviewers' concerns and justify in your rebuttal if you can’t address specific requests experimentally. We cannot make any decision about publication until we have seen the revised manuscript and your response to the reviewers' comments. Your revised manuscript will be sent for further evaluation by the reviewers.

We expect to receive your revised manuscript within 2 months. 

*NOTE: In your point by point response to to the reviewers, please provide the full context of each review. Do not selectively quote paragraphs or sentences to reply to. The entire set of reviewer comments should be present in full and each specific point should be responded to individually, point by point.

To enhance the reproducibility of your results, we recommend that if applicable you deposit your laboratory protocols in protocols.io, where a protocol can be assigned its own identifier (DOI) such that it can be cited independently in the future. For instructions see: https://journals.plos.org/plosbiology/s/submission-guidelines#loc-materials-and-methods. 

Sincerely,

Di Jiang

PLOS Biology

REVIEWS:

Reviewer #1: This is an elegant work by Liu and colleagues that focus in the role of endosomal pathways in EHT. They found Rab5c in an RNA-seq screen of hemogenic specific genes. They have performed many the genetic controls and strategies by using knockdown, dominant negative or constitutively active forms, but also knockout of all Rab5 members (since. Rab5c alone was compensated). They found that depletion of Rab5c leads to a defect in definitive hematopoietic stem and progenitor cells and derivatives. Molecularly they found that Notch pathway is affected by checking the levels of N1IC and a Notch-fluorescent reporter. The levels of Notch are highly regulated by endosomes thus it is not surprising that an early endosome component affects this pathway. However, this is the first time that this specific interaction is reported. 

The effect on the early hematopoietic development of Rab5c is convincing, and the relationship with Notch and AKT as well, however some questions remain about the mechanisms that could be further addressed. 

Does Rab5c co-immunoprecipitates with Notch1 or with Akt? Is this early endosome localization of Notch a required step for activation of Notch, for Notch receptor recycling or for Notch degradation in lysosomes? The authors should address this question more directly. In addition, they only show data of co-localization in cell lines (293T or Hela), could they should some data in the endothelium of zebrafish? Their conclusions are sometimes too strong considering that they never address the endocytic detection in zebrafish. For example in Page 10, line 226: we concluded that Rab5c is essential for HE specification through endocytic trafficking-regulated Notch signaling.

The authors colocalize endosomes with Notch1, Jag1 and Dll4. Are they together in the same endosomes? Is Notch1 co-localizing with any of the ligands? Is the Notch1 antibody is recognizing the intracellular or the extracellular domain? The interpretation would be different depending on what kind of Notch are they detecting.

Specific comments

I have some problems to see the co-localization of Notch members with the endosomal components. In Figure 4 but also other figures, colocalization of notch members with the early endosome marker is not obvious. Could the authors show higher magnification/better resolution of the endosomal images?

The authors should clearly specify in the text all the experiments performed in cell lines, since most of the time it is only in the figure legend.

Do the authors suggest that Notch endosomes are only interacting with EEA1 and LAMP1? Could they show negative controls? they could test colocalization of Notch members with Appl1 and Akt with EEA1. If the authors are correct, those proteins should not co-localize. 

Although authors have checked Wnt and Erk pathways, they cannot exclude that Rab5c is affecting other pathways a part from Notch or Akt/Pi3K. The authors should mention this possibility. 

It is also surprising the DN and CA phenotype is similar in terms of Notch activation. Morover both construts have opposite activity for Rab5c but they seem to affect similarly Notch. The authors should discuss this observation.

Reviewer #2: Paper by Heng et al. provides detailed analysis of Rab5c function during hematopoietic development in zebrafish model. This is a high quality manuscript which revealed that Rab5c-mediated endocytic trafficking regulates Notch signaling to ensure appropriate endosome environment for NICD release during HSPC production. In addition, authors demonstrated that Rab5c activates AKT signaling through interaction with Appl1 adaptor protein which is present in subset of Rab5-positive endosomes. Similar to down-regulation, Rab5c overactivation caused HE specification defect due to ectopic endocytosis of Notch ligands and receptors and increased their transport to endosome-lysosome for degradation. Overall, manuscript established an important role of Rab5c in HSPC development and revealed major molecular mechanisms implicated in Rab5c function in hemogenic endothelium. 

I have just few minor comments: 

1. Western blots in Fig. 3C, 4A, 4G and 6A shows a single experiment. It would be helpful to provide quotative analysis of western blot data from several independent experiments.

2. Spell-out for some abbreviations (WISH and few others) is missing. It would be helpful to spell-out all abbreviations when using for the first time.

Reviewer #3: In this manuscript, Heng and colleagues investigate the role of rab5c in HSPC development in zebrafish. Rab5c was previously identified by the same laboratory in a RNA-seq screen from sorted flk1+runx1- vs. flk1+runx1+ cells to be one of the most up-regulated genes in cells of the hemogenic endothelium. In this study, they use both morpholino oligonucleotides and CRISPR-Cas9 to suppress rab5c expression. They then show by extensive data sets that the gene is essential for the specification of HSPCs by endocytic trafficking of Notch and Akt signaling in zebrafish, which importantly provides new insights in how intrinsic signals are orchestrated to execute cell fate transition. This is interesting work, however some points need to be addressed. 

Unfortunately, the picture quality for the main part of the manuscript was very low making it sometimes hard to interpret the data, but this is most likely because of downsizing by the system, since figures in the supplementary data are much better. Some questions may arise because of this issue. 

Major comments:

It has been previously shown, that Rab5c is a key regulator of trafficking of the CD93/Multimerin/β1-integrin complex in endothelial cell adhesion and migration (Barbera 2019). Furthermore, another recent study demonstrated that another small GTPase of the Ras-related protein family, Rap1b, promotes notch-mediated hemogenic endothelium specification of PLPM cells during HSPC specification by enhancing integrin-mediated cell adhesion. The authors should further investigate and comment on these recent findings in the context of their study. 

The authors use a construct for somite-specific overexpression of rab5c mRNA lacking the MO binding site by the mylz promoter and are not able to observe a rescue of the HSPCS phenotype after rab5c knockdown. From this they claim that the somitic wave of HSPC specification is not involved in this process. They should also consider performing early NICD induction at 14 hpf and analysis of sclerotome markers to further confirm that this wave is not involved. Additionally, they should discuss in the text why they perform these different experiments here (in regard to the two different waves, somitic and endothelial, during HSPC development). 

The authors show rab5c expression in endothelial cells at 26 hpf. If it is indeed responsible for notch pathway activation, it should already be present at earlier time-points between 16 and 20 hpf. Please explain/comment.

Endothelial specific NICD-OE instead of global induction via HS would be better for the Notch rescue experiments in Figure 3 and 6. 

In regard to their apoptosis assays: wouldn't the authors expect to find some double positive (apoptotic and endothelial) cells? What are these apoptotic cells that the authors find in their experiments?

Please mention what the numbers below the WISH pictures are. This could be difficult to understand, especially for an unexperienced reader who is not working in zebrafish.

Please use statistical analyses to investigate whether WISH phenotypes indeed show differences, and quantify immunoblot results.

What about other pathways that regulate similar steps of HSPC development (e.g. Wnt, Erk signaling, Shh, Vegf signaling)? How do they relate to the presented pathway?

Is it possible to analyze endocytic trafficking in vivo in zebrafish instead of using transfected cells? 

Minor comments: 

The AGM region is specific to mammals, the term VDA is more commonly used for zebrafish. 

Please explain the western blot data shown in Supplementary Figure S2B. If wt mRNA was co-injected with the MO, shouldn't there be a rescue of the protein expression and no apparent down-regulation? Is there a specific antibody that can alternatively be used to show that indeed the protein is not there? 

In Supplementary Figure 2SE the mentioned arrows are almost not visible. Please enlarge. 

Why didn’t the authors use their mutants for subsequent analyses? 

Please rearrange Supplementary Figure 2 and 3 to keep the order of appearance of the respective figures in the text.

What does relatively normal mean for their primitive blood phenotypes in Figure S2C? Are primitive myeloid cells affected?

The phenotype in Figure 2M is rather moderate, please consider rephrasing.

The yellow is hard to read in some of the figures (e.g. Figure 5F), consider reorganising.

---

## [Decision Letter · Decision Letter 2]

21 Feb 2020

Dear Dr Liu,

Thank you for submitting your revised Research Article entitled "Rab5c-mediated endocytic trafficking regulates hematopoietic stem and progenitor cell development via Notch and AKT signaling" for publication in PLOS Biology. I have now obtained advice from the original reviewers and have discussed their comments with the Academic Editor. 

We're delighted to let you know that we're now editorially satisfied with your manuscript. However before we can formally accept your paper and consider it "in press", we also need to ensure that your article conforms to our guidelines. A member of our team will be in touch shortly with a set of requests. As we can't proceed until these requirements are met, your swift response will help prevent delays to publication. Please also make sure to address the data and other policy-related requests noted at the end of this email.

*Copyediting*

*Published Peer Review History*

*Early Version*

*Submitting Your Revision*

Sincerely,

Di Jiang

PLOS Biology

DATA POLICY:

Regardless of the method selected, please ensure that you provide the individual numerical values that underlie the summary data displayed in the following figure panels as they are essential for readers to assess your analysis and to reproduce it: 1BC, 2BDEGIKN, 3CEG, 4ABDFHIJLNPRT, 5BDFHKMO, 6CEGI, 7DFGHJLNP, S1B, S2FIOP, S3BG, S4BGJLNP, S5CEHKL, S6BEGIK, S7BDF. 

Reviewer remarks:

Reviewer #1 (Anna Bigas, signed review): The authors have addressed all comments and have considerably improved the manuscript.

Reviewer #2: All my critiques are adequately addressed.

Reviewer #3: The efforts of the authors during the revisions are greatly appreciated. The manuscript has much improved. 

I only have to last minor comments:

Page 10 line 233. Fig. 3O-T should be changed to Fig. 4O-T

Quantification of western blots should be performed throughout the whole manuscript (e.g. also for S5A S5I).

---

## [Editor Report · Decision Letter 3]

24 Mar 2020

Dear Dr Liu,

On behalf of my colleagues and the Academic Editor, Avinash Bhandoola, I am pleased to inform you that we will be delighted to publish your Research Article in PLOS Biology. 

Early Version

PRESS 

Kind regards,

Vita Usova 

Publication Assistant, 

PLOS Biology

on behalf of

Di Jiang,

Associate Editor

PLOS Biology